# The 4DEnVar-based weakly coupled land data assimilation system for E3SM version 2

Pengfei Shi[1], L. Ruby Leung[1], Bin Wang[2], Kai Zhang[1], Samson M. Hagos[1], and Shixuan Zhang[1]

[1]Atmospheric Sciences and Global Change Division, Pacific Northwest National Laboratory, Richland, Washington, USA

[2]State Key Laboratory of Numerical Modeling for Atmospheric Sciences and Geophysical Fluid Dynamics (LASG), Institute of Atmospheric Physics, Chinese Academy of Sciences, Beijing, China

*Correspondence*: L. Ruby Leung (Ruby.Leung@pnnl.gov) and Pengfei Shi (pengfei.shi@pnnl.gov)

**Abstract.** A new weakly coupled land data assimilation (WCLDA) system based on the four-dimensional ensemble variational (4DEnVar) method is developed and applied to the fully coupled Energy Exascale Earth System Model version 2 (E3SMv2). The dimension-reduced projection four-dimensional variational (DRP-4DVar) method is employed to implement 4DVar using the ensemble technique instead of the adjoint technique. With an interest in providing initial conditions for decadal climate predictions, monthly mean anomalies of soil moisture and temperature from the Global Land Data Assimilation System (GLDAS) reanalysis from 1980 to 2016 are assimilated into the land component of E3SMv2 within the coupled modeling framework with a one-month assimilation window. The coupled assimilation experiment is evaluated using multiple metrics, including the cost function, assimilation efficiency index, correlation, root mean square error (RMSE) and bias, and compared with a control simulation without land data assimilation. The WCLDA system yields improved simulation of soil moisture and temperature compared with the control simulation, with improvements found throughout the soil layers and in many regions of the global land. In terms of both soil moisture and temperature, the assimilation experiment outperforms the control simulation with reduced RMSE and higher temporal correlation in many regions, especially in South America, Central Africa, Australia, and large parts of Eurasia. Furthermore, significant improvements are also found in reproducing the time evolution of the 2012 U.S. Midwest drought, highlighting the crucial role of land surface in drought lifecycle. The WCLDA system is intended to be a foundational resource for research to investigate land-derived climate predictability.

## 1 Introduction

The intrinsic chaos of the atmosphere limits traditional weather forecasting to roughly two weeks (Simmons and Hollingsworth, 2002). The feasibility of atmospheric predictability beyond two weeks lies with the interactions of the atmosphere with slowly varying components of the Earth system such as the ocean or land surface, or from predictable external forcings (Guo et al., 2012). Climate prediction can therefore be conceptually divided into both an initial value and a forced boundary value problem (Collins and Allen, 2002; Conil et al., 2007). One of the biggest technical challenges for improving the quality of climate predictions is the initialization of coupled models from observations (Taylor et al., 2012).

Much work has been devoted to initializing climate system models for practicable decadal climate predictions (DCPs). These models couple various components, such as models of the atmosphere, ocean, sea ice, land and river. Due to their complexity, coupled models are often more susceptible to initial conditions (ICs) than their individual model components, underscoring the importance of data assimilation (DA) (Sakaguchi et al., 2012). The application of DA methods is essential to incorporate reanalysis data into the components of coupled model and produce the optimal ICs to improve DCPs. The initialization for DCPs uses both uncoupled DA and coupled data assimilation (CDA) methods. Uncoupled DA performs DA under the framework of an individual component model (e.g., standalone land surface model forced by atmospheric observations or reanalysis data rather than coupled with an atmospheric model), and then the uncoupled DA analyses from different individual components are combined to form the ICs of a coupled model (Zhang et al., 2020). For example, most existing reanalysis data were produced using uncoupled DA approaches, and these reanalysis datasets are then directly used to initialize DCPs in some studies (Du et al., 2012; Bellucci et al., 2013). However, such uncoupled DA often exhibits poor consistency among the ICs of different component models, and eventually produces low prediction skills (Balmaseda et al., 2009; Boer et al., 2016; Ardilouze et al., 2017).

To obtain balanced multi-component ICs in coupled models, recent studies focus on the development of CDA methods under the coupled modeling framework (Penny and Hamill, 2017; He et al., 2020a). The purpose of CDA is to produce balanced and coherent ICs for all components within the climate system by incorporating reanalysis information from one or more components in the coupled model, providing great potential for improving seamless climate predictions (Dee et al., 2014). Some

studies underscore the superior advantages of CDA over traditional uncoupled DA methods (Lea et al.,
2015; Zhang et al., 2005). CDA methods are categorized into two main types: weakly coupled data
assimilation (WCDA) and strongly coupled data assimilation (SCDA). WCDA assimilates the
observations or existing reanalysis into the respective component of the coupled model and then transfers
reanalysis information to the other components through the coupled model integration (He et al., 2020b;
Zhang et al., 2020). Considering that sequential DA encompasses both the analysis and the forecast steps,
WCDA allows no direct influence of reanalysis information from a single component to other
components in the analysis step as the cross-component background error covariances are not used, but
coupling in the forecast step allows interactions across different components during the model integration
(Browne et al., 2019) and propagates reanalysis information to other components. In contrast, SCDA
utilizes cross-component background error covariances to directly assimilate reanalysis information from
one component into all components, treating the entire Earth system model as one unified system (Penny
et al., 2019). Furthermore, similar to WCDA, SCDA also allows coupling in the forecast step to propagate
reanalysis information from one component to the other components (Yoshida and Kalnay, 2018).
Several studies indicate that SCDA typically exhibits more pronounced improvements in assimilation
performance relative to WCDA (Smith et al., 2015; Sluka et al., 2016). However, the application of
SCDA poses substantial technical challenges, particularly in the establishment of effective cross-
component background error covariances. Consequently, the majority of contemporary CDA systems
still utilize the WCDA framework.

Recent research efforts have started to implement the CDA system to initialize DCPs, using a

diverse range of DA techniques from simple to complex. The simplest method is nudging which adjusts
the model states towards the observations or existing reanalysis (Hoke and Anthes, 1976; Zhang et al.,
2020). Although the nudging method is time-saving and easy to implement, its application in CDA is
restricted primarily due to the limited types of observations and the required interpolation of observations
at every time step of model integration (He et al., 2017). Previous studies have developed advanced CDA
systems using variational and filtering approaches, such as the three-dimensional variational data
assimilation (3DVar) (Fujii et al., 2009; Yao et al., 2021), and ensemble-based techniques like the
ensemble Kalman filter (EnKF) (Zhang et al., 2007). The former generally utilizes the stationary

background error covariance and assimilates observations sequentially (Lin et al., 2017). In contrast, the latter uses the flow-dependent forecast error covariance and recursively integrates observations into the model (Lei and Hacker, 2015). Several studies also show encouraging progress in constructing CDA systems using four-dimensional variational data assimilation (4DVar) method (Smith et al., 2015; Fowler and Lawless, 2016). The objective of 4DVar is to optimize four-dimensional model states and provide a compatible temporal trajectory that matches observational records across each assimilation window (Mochizuki et al., 2016). The 4DVar method is an advanced assimilation technique that exhibits superiority over other assimilation techniques like nudging and 3DVar in multiple aspects. Initial shocks that influence prediction skills can be significantly minimized by the 4DVar approach due to the dynamical consistency between the model and ICs (Sugiura et al., 2008). However, it is difficult to apply the 4DVar method for CDA systems in the fully coupled model because of the challenge in adjoint integration of the coupled model and its high computational cost in the analysis step. Finally, to capitalize on the strengths of both ensemble and variational techniques, recent studies focus on developing new hybrid data assimilation methods (Wang et al., 2010; Buehner et al., 2018). The hybrid approach utilizes an ensemble forecast to generate flow-dependent forecast error covariances and presents a way to perform 4DVar optimization without the need for tangent linear and adjoint models (Lorenc et al., 2015). However, most studies on CDA have focused on assimilating observations or reanalysis data of ocean, atmosphere and even sea ice (He et al., 2017; Li et al., 2021; Kimmritz et al., 2018). There have been relatively few instances of CDA studies assimilating land observations or land reanalysis data.

In this study, we introduce the development of the 4DEnVar-based weakly coupled land data assimilation (WCLDA) system for the Energy Exascale Earth System Model version 2 (E3SMv2) (Golaz et al., 2022). The 4DEnVar method in this WCLDA system is the dimension-reduced projection 4DVar (DRP-4DVar; Wang et al., 2010) which utilizes the ensemble technique as an alternative to the adjoint technique for implementing 4DVar. In this WCLDA system, monthly mean anomalies of soil moisture and temperature from a global land reanalysis product are assimilated into the land component of a coupled climate model in the analysis step, and subsequently during the forecast step, the land reanalysis information incorporated into the ICs of the land component is propagated to the other components (e.g., atmosphere and ocean) through the fully coupled model integration and influences the ICs of all

components for the next assimilation window. The primary goal of the WCLDA system is intended to be
a foundational resource for exploring predictability of the Earth system by the E3SM community,
specifically focusing on understanding the sources of predictability provided by land versus ocean, with
an initial focus on DCPs. This WCLDA system also provides the groundwork for future actionable
predictions of Earth system variability using E3SM.

The objective of this paper is to introduce the implementation of the 4DEnVar-based WCLDA

system for the land component of E3SMv2. In Section 2, we provide an overview of the E3SMv2 model,
describe the 4DEnVar methodology in detail and outline the framework of the 4DEnVar-based WCLDA
system. Preliminary evaluation of the WCLDA system is presented in Section 3. Finally, conclusions are
discussed in Section 4.

**2 Methods**
**2.1 Model Description**

The model used in this study is a relatively new state-of-the-art Earth system model known as

Energy Exascale Earth System Model version 2 (E3SMv2), supported by the U.S. Department of Energy
(DOE) to improve actionable Earth system predictions and projections (Leung et al., 2020). The
atmospheric component is the E3SM Atmosphere Model version 2 (EAMv2), which is built on the
spectral-element atmospheric dynamical core with 72 vertical levels (Dennis et al., 2012; Taylor et al.,
2020). At the standard resolution, EAMv2 is applied on a cubed sphere with a grid spacing of ~100 km
for the dynamics. The ocean component is the Model for Prediction Across Scales-Ocean (MPAS-O),
which applies the underlying spatial discretization to the primitive equations with 60 layers using a z-
star vertical coordinate (Petersen et al., 2018; Reckinger et al., 2015). The sea ice component is MPAS-
SI, which shares the same Voronoi mesh with MPAS-O, with mesh spacing varying between 60km in the
mid-latitudes and 30 km at the equator and poles (Golaz et al., 2022). The land component is the E3SM
Land Model version 2 (ELMv2), which is based on the Community Land Model version 4.5 (CLM4.5)
(Oleson et al. 2013). Simulations are run in a satellite phenology mode with prescribed leaf area index,
and the prescribed vegetation distribution has been updated for better consistency between land use and
changes in plant functional types described by Golaz et al. (2022). The river transport component is the
Model for Scale Adaptive River Transport version 2 (MOSARTv2), which provides detailed
representation of riverine hydrologic variables (Li et al., 2013). These five components exchange fluxes
through the top-level coupling driver version 7 (CPL7) (Craig et al., 2012). Further details on the
E3SMv2 model are described in Golaz et al. (2022).

**2.2 Datasets**
Monthly mean soil moisture and soil temperature data in a total of ten soil layers are produced by
the Global Land Data Assimilation System (GLDAS; Rodell et al., 2004). The GLDAS product generates
optimal fields of land surface states and fluxes in near-real time by forcing multiple offline land surface
models with observation-based data fields. These reliable and high-resolution global land surface datasets
from GLDAS are extensively utilized in weather and climate studies, hydrometeorological investigations
and water cycle research (Chen et al., 2021; Zhang et al., 2018). The GLDAS datasets have been available
globally at high spatial resolution since January 1979 and can be accessed through the Goddard Earth
Science Data and Information Service Center. For more consistency with ELMv2, we utilize GLDAS
data produced by CLM. In contrast to decadal timescales, data signals with temporal resolutions shorter
than one month can potentially introduce undesirable noise, which can adversely affect DCPs when high
temporal resolution data are assimilated into the ICs. Moreover, it is very computationally demanding to
assimilate complex actual observations in the initialization for DCPs that requires long-term DA cycles.
Therefore, similar to most existing initialization approaches for DCPs that assimilate reanalysis data, this
study describes the implementation of a data assimilation approach for initializing DCPs by assimilating
monthly mean GLDAS data within the one-month assimilation window.
Monthly mean surface soil moisture data from the Advanced Microwave Scanning Radiometer
(AMSR) and land surface temperature data from the Moderate Resolution Imaging Spectrometer
(MODIS) are utilized for validation. (1) The AMSR data provides surface soil moisture estimations by
measuring the microwave emission from the Earth's surface (Njoku et al., 2003). The soil moisture data
from AMSR are widely used in scientific research to study surface water cycles, drought conditions and
hydrologic modeling (Du et al., 2019; McCabe et al., 2008). (2) MODIS is an essential instrument
onboard the Terra and Aqua satellite platforms (Remer et al., 2005). The MODIS datasets provide
comprehensive global observations describing atmospheric, terrestrial and oceanic conditions, including
land surface temperature, vegetation indices and cloud properties (Justice et al., 2002). The MODIS
products are extensively utilized for monitoring environmental changes and supporting climate change
research (Gao et al., 2015; Mertes et al., 2015).

Current initialization techniques are broadly classified into two categories: full-field assimilation

with reanalysis values, and anomaly assimilation with reanalysis anomalies (Hu et al., 2020; Polkova et
al., 2019). The full-field assimilation is commonly performed to reduce the influence of systematic model
biases by replacing the initial model state with the optimal available estimate of the reanalysis state (Volpi
et al., 2017). However, the model trajectory tends to drift away from the observations and revert to the
model's inherent preferred state because of model deficiencies (Smith et al., 2013). This problem is
partially addressed with the anomaly assimilation by assimilating the reanalysis anomalies added to the
model climatology (Carrassi et al., 2014). In this study, we conduct the anomaly assimilation for the
WCLDA system with bias correction applied to GLDAS data before assimilation. For bias correction,
the difference between GLDAS data and its long-term average is calculated as anomalies and then added
to the simulated model climatology.

**2.3 Data Assimilation Scheme**

The 4DEnVar algorithm in this study is based on the DRP-4DVar technique, which is an efficient

pathway for applying 4DVar through using the ensemble method rather than the adjoint technique (Wang
et al., 2010). The DRP-4DVar method generates the optimal estimation in the sample space through
aligning the observations with ensemble samples along the coupled model trajectory (Liu et al., 2011).

DRP-4DVar is an economical approach that minimizes the cost function of the standard 4DVar by

using the ensemble technique instead of the adjoint technique (Wang et al., 2010). The background error
covariance matrix $B$ is estimated using the pure ensemble covariance. The ensemble members originate
from historical or ensemble forecasts. Considering the high computational cost of ensemble forecasts for
the coupled model in our study, we utilize outputs from the pre-industrial control (PI-CTRL) experiment
of E3SMv2 to generate ensemble members. The instantaneous state at the beginning of each month and
the corresponding monthly mean state of this month from the 100-year balanced PI-CTRL simulation
are used as the samples of initial condition $(x_i)$ and forecast samples $(y_i)$. The corresponding perturbation
samples are calculated as $x_i' = x_i - \bar{x}$ and $y_i' = y_i - \bar{y}$, where $\bar{x}$ and $\bar{y}$ are the 100-year average
values of $x_i$ and $y_i$ at the same month, respectively. Then, $m$ pairs of perturbation samples
$(x_1', x_2', x_3', \cdots, x_m')$ and $(y_1', y_2', y_3', \cdots, y_m')$ are selected at each DA analysis step according to the
correlations between $y_i'$ and the observational innovation $y_{obs}' = y_{obs} - y_b$, ensuring that each $y'$
sample is selected independently of the other samples in the ensemble. In this study, $m = 30$. Then the
estimation of the background error covariance matrix $B$ is represented by the formula in Eq. (1), utilizing
the selected $x'$ samples. We implement both horizontal and vertical localization to reduce sampling
errors due to the finite ensemble size and to alleviate the spurious remote influence from distant grid
points. Our approach to horizontal localization is to apply a distance-dependent weighting function to
the background error covariance. The vertical localization is employed to limit the influence of reanalysis
information on specific soil layers. Please refer to Wang et al. (2018) for more detailed descriptions of
the localization methodology in our study.

$$
\begin{cases}
B = bb^T \\
b = \dfrac{1}{\sqrt{m-1}} \times (x_1' - \bar{x}', x_2' - \bar{x}', x_3' - \bar{x}', \cdots, x_m' - \bar{x}') \\
\bar{x}' = \dfrac{1}{m}(x_1' + x_2' + x_3' + \cdots + x_m')
\end{cases} \quad (1)
$$


According to Wang et al. (2010), DRP-4DVar produces the analysis increment $(x_a')$ by minimizing
the 4DVar cost function in the incremental form (Courtier et al., 1994):

$$
\begin{cases}
J(x_a') = \min_{x'} J(x') \\
J(x') = \frac{1}{2}(x')^T B^{-1} x' + \frac{1}{2}(\tilde{y}' - \tilde{y}_{obs}')^T (\tilde{y}' - \tilde{y}_{obs}')
\end{cases} \quad (2)
$$


Here $x' = x - x_b$ represents the increment of model variables relative to the background; $\tilde{y}_{obs}' =$
$r^{-1} y_{obs}' = r^{-1}(y_{obs} - y_b)$ denotes the weighted observational innovation for monthly mean anomalies
of soil moisture and temperature, and $R = rr^T$ is the observational error covariance matrix that is
usually diagonal; $\tilde{y}' = r^{-1} y' = r^{-1}(y - y_b)$ is the weighted projection of the increment $(x')$ onto the
observation space; the superscript $T$ represents the transpose.
To simplify the calculation of the minimization, the increment of model state variables $x'$ and the
corresponding weighted observation increment $\tilde{y}'$ are projected onto the dimension-reduced sample
space through the following projection transformations:
$$\begin{cases} x' = P_x\alpha \\ \tilde{y}' = P_y\alpha \end{cases} \tag{3}$$

where $\alpha$ is the $m$-dimension column vector containing the weight coefficients $(\alpha_1, \alpha_2, \alpha_3, \cdots, \alpha_m)$; $P_x$
and $P_y$ denote the projection matrices that incorporate the initial condition perturbations and the
corresponding monthly mean samples as follows:
$$\begin{cases} P_x = (x'_1, x'_2, x'_3, \cdots, x'_m) \\ P_y = (\tilde{y}'_1, \tilde{y}'_2, \tilde{y}'_3, \cdots, \tilde{y}'_m) \end{cases} \tag{4}$$

where $\tilde{y}'_i = r^{-1}y'_i$ $(i = 1, 2, \cdots, m)$. Then the original 4DVar cost function defined in Eq. (2) is
transformed into the following new cost function and the analysis can be computed in the sample space
by minimizing this new cost function:
$$\begin{cases} \tilde{J}(\alpha_a) = \min_\alpha \tilde{J}(\alpha) \\ \tilde{J}(\alpha) = \frac{1}{2}\alpha^T B_\alpha^{-1}\alpha + \frac{1}{2}(P_y\alpha - \tilde{y}'_{obs})^T(P_y\alpha - \tilde{y}'_{obs}) \\ x_a = x_b + x'_a = x_b + P_x\alpha_a \end{cases} \tag{5}$$

The solution to this minimization problem is formulated as:
$$\alpha_a = (B_\alpha^{-1} + P_y^T P_y)^{-1} P_y^T \tilde{y}'_{obs} \tag{6}$$

In this study, the DRP-4DVar-based WCLDA system is used to incorporate the land reanalysis data only.
The optimal analysis for the land state variables $(x_a^{lnd})$ is obtained by adding the analysis increment
$(x'^{lnd}_a)$ to the background of land ICs $(x_b^{lnd})$, as expressed in Eq. (7):
$$x_a^{lnd} = x_b^{lnd} + x'^{lnd}_a = x_b^{lnd} + P_x(B_\alpha^{-1} + P_y^T P_y)^{-1} P_y^T \tilde{y}'_{obs} \tag{7}$$

In the analysis step, only the land state variables are updated to the optimal analysis ($x_a^{lnd}$).
Subsequently, we proceed with a one-month freely coupled integration of the E3SMv2 model during the
forecast step. This integration is initialized from the optimal land ICs ($x_a^{lnd}$) along with the background
fields as the ICs of other components (e.g., atmosphere and ocean). Throughout this one-month free
integration, the interactions among the model components indirectly enhance the background states of
these components (e.g., atmosphere and ocean) for the next assimilation window due to the more realistic
land state variables. Moreover, this coupled integration also contributes to the balance between the ICs
of different components.

**2.4 4DEnVar-based WCLDA System**
The 4DEnVar-based WCLDA system is developed to assimilate the monthly mean soil moisture and
temperature data from the GLDAS analysis dataset into the land component of E3SMv2 using the DRP-
4DVar method. Two sets of numerical experiments are conducted to evaluate the performance of land
data assimilation in the WCLDA system. The control simulation (CTRL) is a 37-year freely coupled
integration driven by observed external forcings (e.g., solar radiation, greenhouse gas and aerosol
concentrations) from 1980 to 2016. In the freely coupled simulation, the various components of the Earth
system model, namely the atmosphere, land, river, ocean, and sea ice, interact dynamically without any
constraints. The observed external forcing mainly acts on the atmospheric component and then influences
other components (e.g., land surface, ocean, and sea ice) through their coupling with the atmosphere.
CTRL provides the benchmark for assessing the performance of the WCLDA system. The assimilation
experiment (Assim) is conducted from 1980 to 2016 based on the WCLDA system in which the GLDAS
data are assimilated into the land state variables from the first to the tenth layer with a one-month
assimilation window under the coupled modeling framework. The effectiveness of the WCLDA system
is evaluated through the comparison between Assim and CTRL. In both Assim and CTRL, the transient-
historical external forcings are prescribed following the CMIP6 protocol (Eyring et al., 2016).

The flowchart of the 4DEnVar-based WCLDA system is illustrated in Figure 1. The DRP-4DVar

method incorporates three inputs: model background, observational innovation and 30 perturbation
samples. First, the E3SMv2 model is executed for one month, during which state variables such as model
background ($x_b$), observational operator ($H$) and observational background ($y_b$) are stored. The model
background ($x_b$) denotes the monthly initial states before assimilation, and the observational operator ($H$)
represents a one-month integration by the coupled model to generate monthly mean model outputs ($y_b$).
Second, upon completion of the one-month coupled run, the observational innovation ($\tilde{y}'_{obs}$) is determined
by calculating the differences in soil moisture and temperature between the monthly mean GLDAS data
($y_{obs}$) and the model outputs ($y_b$). From the 100-year sample database of the E3SMv2 PI-CTRL
simulation, 30 samples of monthly mean perturbation ($\tilde{y}'$) are chosen with the highest absolute correlation
with the observational innovation. The corresponding 30 monthly IC samples ($x'$) are also obtained.
Finally, the analysis increment is generated in the sample space and the optimal analysis ($x_a$) is calculated
using the DRP-4DVar algorithm.

The schematic diagram in Figure 2 outlines the assimilation process of the 4DEnVar-based WCLDA

280 system in E3SMv2. The incorporation of GLDAS data into the E3SMv2 model consists of the analysis

281 step and the forecast step. In the analysis step, the differences between monthly mean GLDAS data and

282 model outputs are calculated and utilized to produce the optimal assimilation analysis at the beginning of

283 a one-month assimilation window. Subsequently, in the forecast step, this optimal assimilation analysis is

284 used as the land ICs combined with the background ICs for other components to conduct one-month

285 forecast using the E3SMv2 model. This forecast generates the backgrounds of all model components for

286 the next assimilation window. As a result, the forecasted backgrounds for all components are influenced

287 by the land reanalysis information incorporated into the ICs of the land component. In general, when the

288 coupled model is used in the forecast step while the optimal assimilation analysis is updated separately

289 for the respective component, the assimilation approach is identified as WCDA (Penny et al., 2019; Zhang

290 et al., 2020).

291   The detailed assimilation process mainly consists of three steps within each one-month assimilation

292 window: 1) the E3SMv2 model is initially executed for one month to generate the simulated monthly

293 mean soil moisture and temperature ($y_b^{lnd}$); 2) the observational innovation ($y_{obs}'$) is obtained through

294 subtracting the model simulation ($y_b^{lnd}$) from the monthly mean observation ($y_{obs}^{lnd}$). This innovation is

295 then applied to formulate the optimal assimilation analysis of land surface ($x_a^{lnd}$) at the beginning of the

296 assimilation window through the DRP-4DVar method; 3) the E3SMv2 model is rewound to the start of

297 the month and the second one-month model run is executed using the optimal ICs ($x_a$) to generate the

298 background for the next assimilation window. Due to multi-component interactions during the one-month

299 freely coupled integration, the land reanalysis information can potentially benefit other components (e.g.,

300 atmosphere and ocean) in the coupled modeling framework (Li et al., 2021; Shi et al., 2022). To assimilate

301 the monthly mean GLDAS product, fully coupled integration by the E3SMv2 model is performed twice

302 within each one-month assimilation window: first to generate the observational innovation by computing

303 the differences between the GLDAS data and model outputs for analysis, and second to forecast the

304 backgrounds of all components for the next assimilation window. When the fully coupled model is

305 executed for the second one-month run, the land reanalysis information is transferred to the other

306 components through multi-component interactions. This approach is similar to previous studies that

307 employed the "two-step" scheme in which the land model integration is performed twice within the same

month to assimilate the monthly GRACE-based TWS observations (Houborg et al., 2012; Girotto et al.,

2016).


**2.5 Evaluation Metrics**

The reduction rate of the cost function is a significant metric for verifying the effectiveness of the

WCLDA system and evaluating the extent of reanalysis information assimilated by the coupled model,
which is formulated as:
$$\begin{cases} \frac{J_1 - J_0}{J_0} \times 100\% \\ J_0 = \frac{1}{2}(y_{obs} - y_b)^T R^{-1}(y_{obs} - y_b) \\ J_1 = \frac{1}{2}(y_{obs} - y_a)^T R^{-1}(y_{obs} - y_a) \end{cases} \tag{8}$$

where $J_0$ and $J_1$ denote the cost function before and after assimilation respectively, $y_{obs}$ represents the
GLDAS data, $y_a$ denotes the monthly mean analyses, $y_b$ is the observation-space background, and $R$ is
defined as the observation error covariance matrix. The observation error covariance matrix $R$ can be
determined statistically by estimating the variance and covariance of the GLDAS data. Negative value
for this metric indicates that reanalysis information has been correctly incorporated into the model
variables.

Following Yin et al. (2014), the assimilation efficiency (AE) index is defined to evaluate the efficiency

of the WCLDA system as follows:
$$AE = \frac{RMSE_{Assim}}{RMSE_{CTRL}} - 1 \tag{9}$$

In this equation, $RMSE_{Assim}$ is the root mean square error (RMSE) between the analysis value from
Assim and the reference data, while $RMSE_{CTRL}$ represents the RMSE between CTRL and the reference
data. Negative (positive) AE value indicates improvements (degradations) by the assimilation. In the
following sections, we use the GLDAS data as the main reference data to verify the correctness of the
WCLDA system, but some analyses are also performed using AMSR surface soil moisture and MODIS
land surface temperature as the reference data.

**3 Results**
**3.1 Evaluation of the cost function**
Figure 3 displays the time series of the monthly reduction rate of the cost function in the 4DEnVar-
based WCLDA system. In the first month, the reduction rate reaches approximately 26.06% in Assim.
Over the subsequent months, Assim maintains the average reduction rate of 7.73% throughout the entire
37-year period. Furthermore, negative reduction rates are observed in 98.65% of the total months,
indicating the effectiveness of the WCLDA system. These results suggest that the WCLDA system is
correctly implemented, with GLDAS data successfully assimilated into the coupled model.

**3.2 Evaluation of the AE index**
The spatial pattern of the AE index for soil moisture at different depths is depicted in Figure 4. The
AE value exhibits negative signal in most areas for total ten soil layers, suggesting the reduction in RMSE
for soil moisture after assimilation. Significant improvements appear over North America, South America,
southern Africa, Europe, and Asia. However, assimilation performance is degraded in the northern part of
Russia and northern Africa. This is consistent with the findings in other studies that assimilation updates
in northern Russia are limited due to the complexities of accurately representing frozen ground and snow
processes in high latitudes (Edwards et al., 2007; Ireson et al., 2013). As surface soil moisture is highly
susceptible to atmospheric conditions, assimilation performance of surface soil moisture is limited by the
accuracy of atmospheric forcing. Furthermore, some degradations found in the deep layers could be
attributed to the substantial influence of various terrestrial factors, such as subsurface runoff and
interactions with groundwater, similar to the findings in previous studies (Liu and Mishra, 2017; Zeng
and Decker, 2009).
Figure 5 shows the spatial distribution of the AE index for soil temperature from surface to deep
layers. Most grid cells from the ten soil layers are dominated by negative AE signals, indicating improved
performance for soil temperature after assimilation. Moreover, the spatial patterns across different soil
layers are highly consistent with each other and exhibit similar magnitudes in most areas. Notable
improvements are observed in central Europe, South America, eastern Russia, and large parts of Eurasia
and North America. In contrast, slight degradations appear over Southeast Asia and along the northern
fringes of Africa. This may be partly related to model uncertainties and possible atmospheric noise, as
shown by many past studies (Kwon et al., 2016; Lin et al., 2020).
We further perform an analysis of the spatial pattern of the AE index for surface soil moisture and
land surface temperature between satellite data and model simulations (Figure A1). For surface soil
moisture, the comparison with AMSR data suggests that the majority of global regions exhibit reduced
RMSE after assimilation. The reduction of RMSE is pronounced in central North America, South America,
southern Africa, Australia, and Europe. However, in high-latitude areas, significant degradations are
observed in northern Russia, which may be possibly related to model deficiencies in simulating the
complex frozen ground and snow processes (Edwards et al., 2007; Ireson et al., 2013). Regarding land
surface temperature, improved performances are evident over South America, Australia, southern Africa,
and large parts of Eurasia when compared to MODIS data. In contrast, some degradations appear over
parts of North America and central Asia, which still require further improvement.

**3.3 Evaluation of the correlation**
Figure 6 displays the spatial patterns of the differences in temporal correlations for soil moisture
between Assim and CTRL with GLDAS data across different soil layers. The majority of global regions
in Assim exhibit higher correlations from the first to the tenth layer compared with CTRL, suggesting the
overall good performance of the WCLDA system. Enhanced correlations in deep soil layers are more
pronounced than in shallow layers, which may be attributed to the longer memory of soil processes in the
deeper soil layers (Wang et al., 2010). Improved correlations appear over North America, central Europe,
Asia, and parts of Africa. However, some scattered areas show slight degradations, such as northern South
America, central Africa, and eastern Russia. Overall, Assim outperforms CTRL with higher correlation
(Figure 6) and lower RMSE (Figure 4) in many regions, such as Europe, North America, southern South
America, and South Asia.
The correlation differences in soil temperature between Assim and CTRL from surface to deep
layers are shown in Figure 7. Assim yields improved correlations from the first to the tenth layer across
the majority of global regions. Furthermore, similar spatial patterns and magnitudes are observed in the
performance of different soil layers, implying the significant heat transfer from the surface to deep zone
that constrains soil temperature across the soil column. Notable improvements are located over South
America, central Africa, Australia, central Europe, and East Asia. Nevertheless, some degradations

appear over North America, western Europe, and Northeast China. Assim shows superior performance over CTRL for soil temperature with higher correlation (Figure 7) and lower RMSE (Figure 5) in many regions, including South America, central Europe, Australia, and central Africa.

**3.4 RMSE and bias of the global mean soil moisture and temperature**

The vertical distributions of RMSE differences between Assim and CTRL for soil moisture and temperature are evaluated in Figure 8. Compared with CTRL, Assim shows noticeable improvements with reduced RMSE for both soil moisture and temperature in all ten soil layers. For soil moisture, the reduction of RMSE increases with depth from the upper to deep soil layers, reaching its maximum at the tenth layer. This could be attributed to the longer soil memory in deep layers than shallow layers. For soil temperature, the reduction of RMSE exhibits similar magnitude from the surface to deep soil layers, which may be explained by the significant heat transfer across different soil layers in regulating soil temperature throughout the soil column.

Figure 9 shows the time evolutions of the vertically averaged global mean soil moisture and temperature bias and RMSE differences. For soil moisture bias (Figure 9a), CTRL exhibits dry biases during the first twenty years and wet biases afterwards. In contrast, Assim shows smaller biases during both periods by reducing the dry bias prior to ~2000 and the wet bias thereafter. Assim also exhibits reduced RMSE (Figure 9b) for soil moisture throughout the entire 37-year period. For soil temperature bias (Figure 9c), CTRL and Assim display comparable performances, possibly due to the small magnitude of model deviation in soil temperature. The RMSE differences (Figure 9d) suggest that Assim decreases the RMSE for soil temperature in the majority of months, with 74.10% of the total months in Assim exhibiting lower RMSE than CTRL. In summary, the superior performance for both soil moisture and temperature in Assim demonstrates that land reanalysis information has been effectively incorporated into the model variables through the WCLDA system.

Noticeably, the simulated soil temperature and soil moisture display similar long-term trends, with cold and dry biases before ~2000 and warm and wet biases afterwards. The soil temperature biases may be related to the global surface air temperature simulated in E3SMv2, which is notably too cold compared to the observed record during the 1970s and 1980s while the model warms up quickly after ~year 2000

(see Figure 23 of Golaz et al., 2022). The global surface air temperature biases during the past decades in
E3SMv1 and v2 have been attributed to the strong aerosol forcing in the model (Golaz et al., 2019; 2022).
As the global mean precipitation scales with the surface temperature at ~2% per degree (Allen and Ingram,
2002), model biases in surface temperature are reflected in biases in precipitation and hence soil moisture,
resulting in similar long-term trends between soil temperature and soil moisture biases in the simulations.

**3.5 2012 U.S. Midwest Drought**
To further evaluate the performance of the WCLDA system, we briefly investigate the impact of land
data assimilation on simulating the temporal evolution of the U.S. Midwest drought in 2012. Time series
of soil moisture percentiles over the Midwest ($36°$-$50°$N, $102°$-$88°$W) demonstrate significant
improvements by Assim in reproducing the time evolution of agricultural drought in 2012 compared with
CTRL (Figure 10). From ERA-Interim data, the agricultural drought starts in August 2011, follows by a
brief relief in early spring of 2012, peaks in September 2012, and recovers by January 2013. The drought
develops rapidly between May and July 2012 over a wide-spread area including the central and
midwestern U.S. This flash drought caused significant agricultural damages and economic losses.
The free running CTRL experiment fails to simulate the temporal evolution of the 2012 Midwest
drought, with a correlation coefficient between CTRL and ERA-Interim of only 0.27. The onset and peak
of the drought are remarkably well captured by Assim, although the drought recovery occurs two months
later than observed. The correlation coefficient of the Assim time series with ERA-Interim is 0.56, which
is statistically significant at the 95% confidence level. Our results highlight the importance of land surface
states for drought lifecycle, with the potential to improve future drought predictions through the
implementation of the WCLDA system.

**4 Conclusions**
In this study, we developed the 4DEnVar-based WCLDA system for the E3SMv2 model and
evaluated the performance of this WCLDA system. The DRP-4DVar method was employed for
implementing 4DVar using the ensemble method rather than the adjoint technique. Special attention is
paid to directly assimilating monthly mean land reanalysis data in this system without interpolating to
every time step. Within each one-month assimilation window, we assimilate land reanalysis information
into the coupled model without breaking the land-atmosphere interaction, which is important for the
WCLDA system to be used to understand the potential sources of predictability provided by land.
Monthly mean anomalies of soil moisture and temperature from the GLDAS reanalysis are
assimilated from 1980 to 2016 through the WCLDA system, and its performance is evaluated using
multiple metrics, including the cost function, AE index, correlation, RMSE and bias. Compared with
CTRL, the cost function is reduced by Assim in most months, suggesting that the GLDAS reanalysis data
has been effectively incorporated into the model. In terms of both soil moisture and temperature, Assim
outperforms CTRL with lower RMSE and higher temporal correlation in many regions, especially in
South America, central Africa, Australia, and large parts of Eurasia. For soil moisture bias, Assim further
decreases the dry bias during the first twenty years and the wet bias thereafter. It is noteworthy that the
subseasonal-to-seasonal time evolution of soil moisture percentiles during the 2012 U.S. Midwest drought
can be quite well captured in Assim, underscoring the significant role of land surface states in drought
propagation.
Our current WCLDA system has some limitations and requires future improvements. Future
enhancements of our WCLDA system will explore the assimilation of additional land products,
particularly those derived from satellite observations. The incorporation of such satellite-based datasets
is expected to further improve the performance of the WCLDA system. It is possible that the influence of
the WCLDA system on atmospheric processes may be limited in some domains due to uncertainties of
the model parameterizations, particularly in representing land-atmosphere interactions (Zhou et al., 2023).
For example, in humid regions where the evaporation process is predominantly energy-limited, the
assimilation of soil moisture tends to exert limited influence. Instead, the assimilation of soil temperature
may yield more substantial improvements. This underscores the importance of the unique characteristics
and constraints presented by complicated regional conditions in the application of assimilation processes.
To this end, the application of the WCLDA system would motivate future work to better understand the
roles of the land surface in climate variability and provide a foundational resource for future predictability
studies by the E3SM community.

*Code and data availability.* The E3SMv2 source codes used in this study can be accessed on Zenodo at
https://zenodo.org/record/8194050. The GLDAS monthly soil moisture and soil temperature data are
available online (https://disc.gsfc.nasa.gov/datasets?keywords=GLDAS%20monthly&page=1). The
MODIS monthly land surface temperature data can be downloaded from the website
(https://disc.gsfc.nasa.gov/datasets/MOD11CM1D_005/summary). The AMSR monthly surface soil
moisture data are available from https://doi.org/10.11888/Soil.tpdc.270960. The ERA-Interim monthly
soil moisture data are available at https://apps.ecmwf.int/archive-
catalogue/?levtype=sfc&type=an&class=ei&stream=moda&expver=1. The model data used in this study
can be found on Zenodo at https://zenodo.org/record/8148737.

*Author contributions.* LRL initiated this study. PS and LRL designed the experiments. PS developed the
data assimilation code and performed the simulations. BW provided advice on the data assimilation
technique and KZ and SZ provided assistance with the E3SM model. PS and LRL analyzed and
interpreted the data. PS and LRL wrote the paper. BW, KZ, SMH, and SZ contributed to the revision.

*Competing interests.* The authors declare no competing interests.

*Acknowledgements*. This research was supported by the Office of Science, Department of Energy
Biological and Environmental Research as part of the Regional and Global Model Analysis program area.
Pacific Northwest National Laboratory is operated by Battelle Memorial Institute for the U.S.
Department of Energy under contract DE-AC05-76RL01830.

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

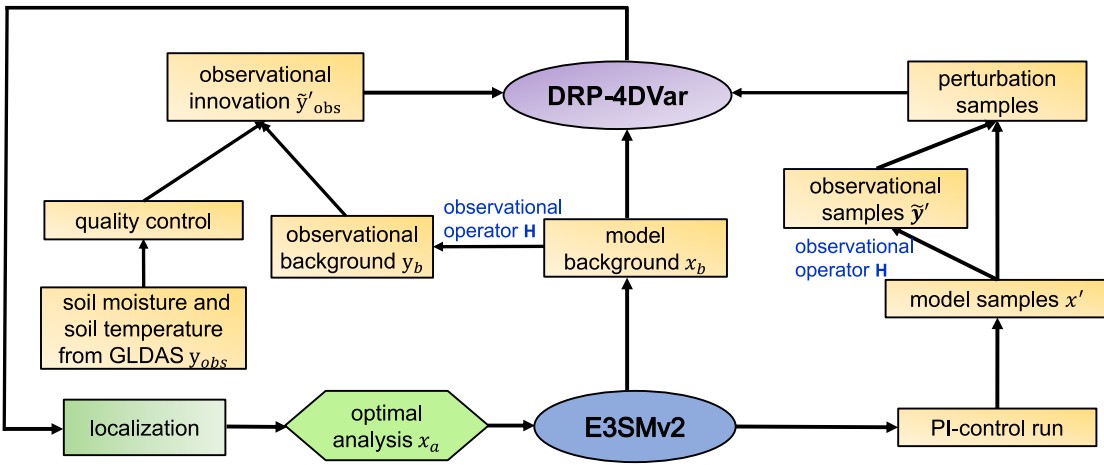


**Figure 1.** Flowchart of the 4DEnVar-based WCLDA system in E3SMv2 based on the DRP-4DVar
method.

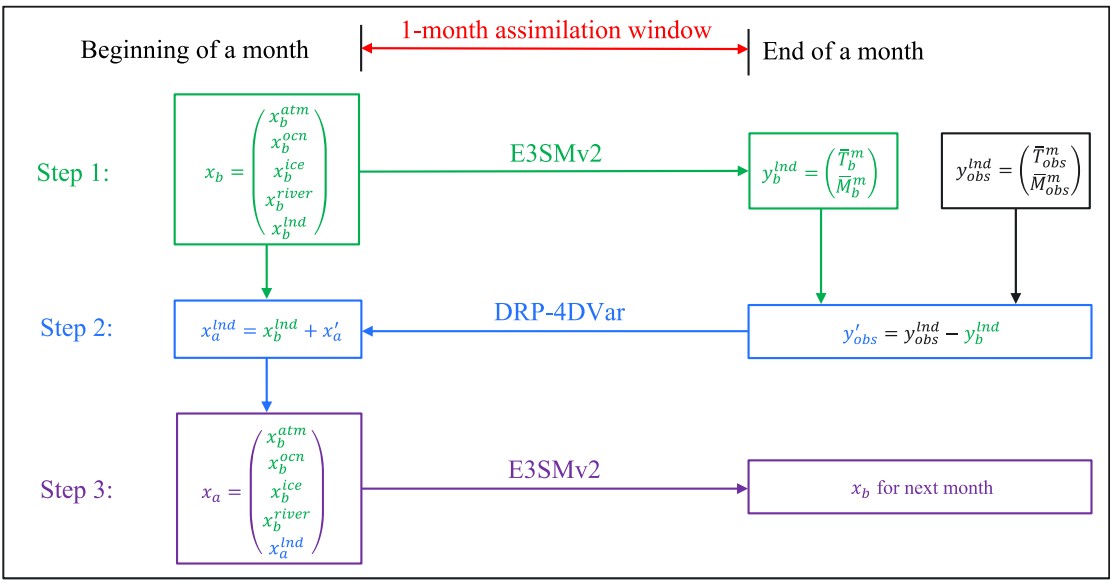

**Figure 2.** Schematic flowchart of the 4DEnVar-based WCLDA system. The beginning of a month is at 0000 UTC on the first day of the month, and the end of the month is at 0000 UTC on the first day of the next month. $x_b$ denotes the background vector including the backgrounds of all E3SMv2 components (atmosphere ($x_b^{atm}$), ocean ($x_b^{ocn}$), sea ice ($x_b^{ice}$), river transport ($x_b^{river}$) and land surface ($x_b^{lnd}$)). $x_a$ consists of the assimilation analysis of land surface ($x_a^{lnd}$) and the backgrounds of other components. $y_b^{lnd}$ represents the simulated monthly mean soil temperature ($\bar{T}_b^m$) and moisture ($\bar{M}_b^m$) by E3SMv2 using $x_b$ as the initial condition. $y_{obs}^{lnd}$ denotes the monthly mean GLDAS data of soil temperature ($\bar{T}_{obs}^m$) and moisture ($\bar{M}_{obs}^m$). $y_{obs}'$ denotes the observational innovation, which is the difference between the GLDAS data ($y_{obs}^{lnd}$) and the observational background ($y_b^{lnd}$).

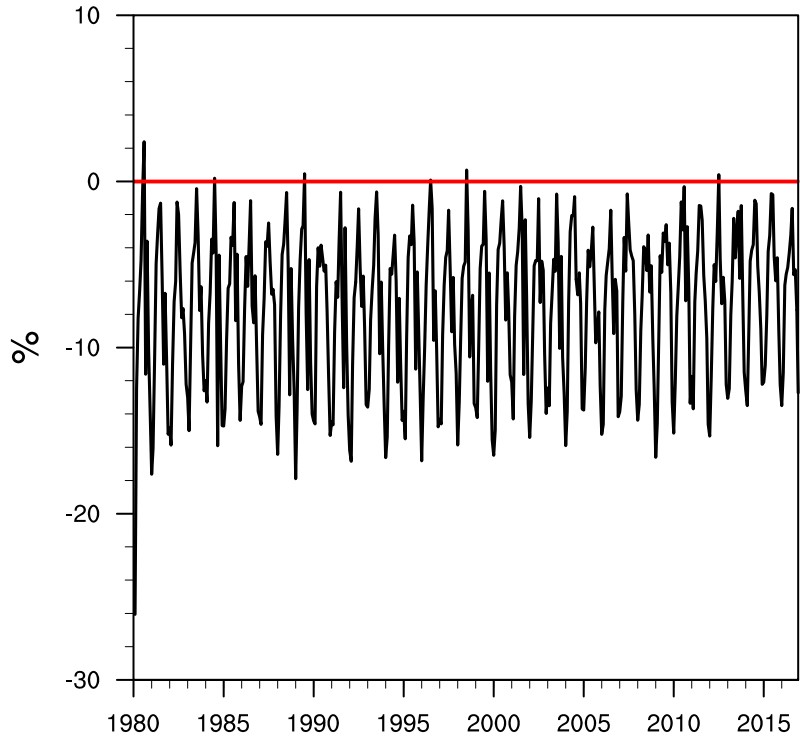


**Figure 3.** Time series of the reduction rate of the cost function from 1980 to 2016 in the 4DEnVar-based
WCLDA system.

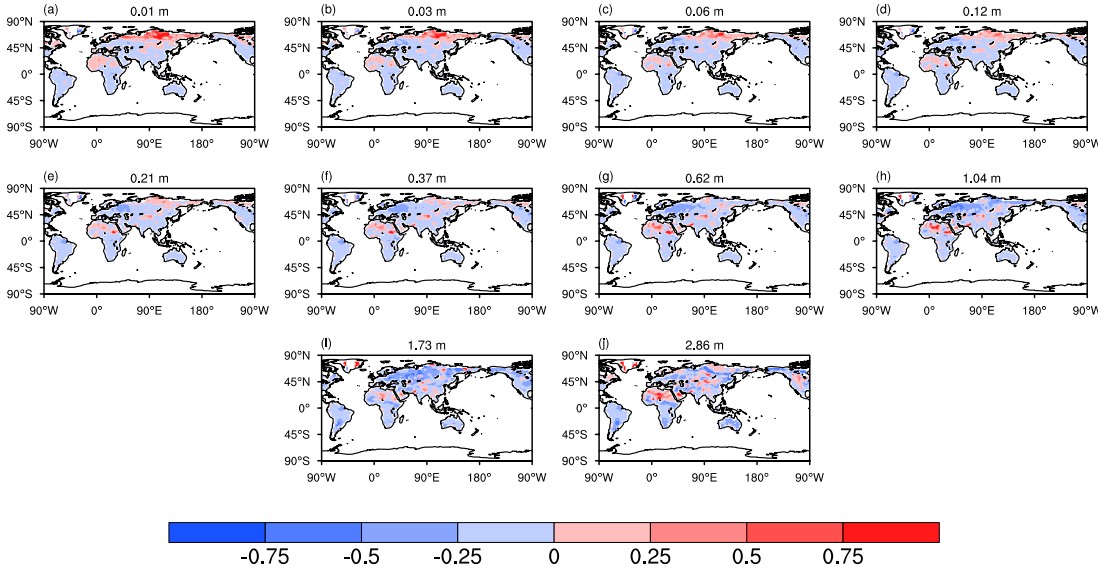

**Figure 4.** Spatial distribution of the AE index for soil moisture from the surface to deep layers during the 1980-2016 period. The number at the top center denotes the depth of each soil layer.

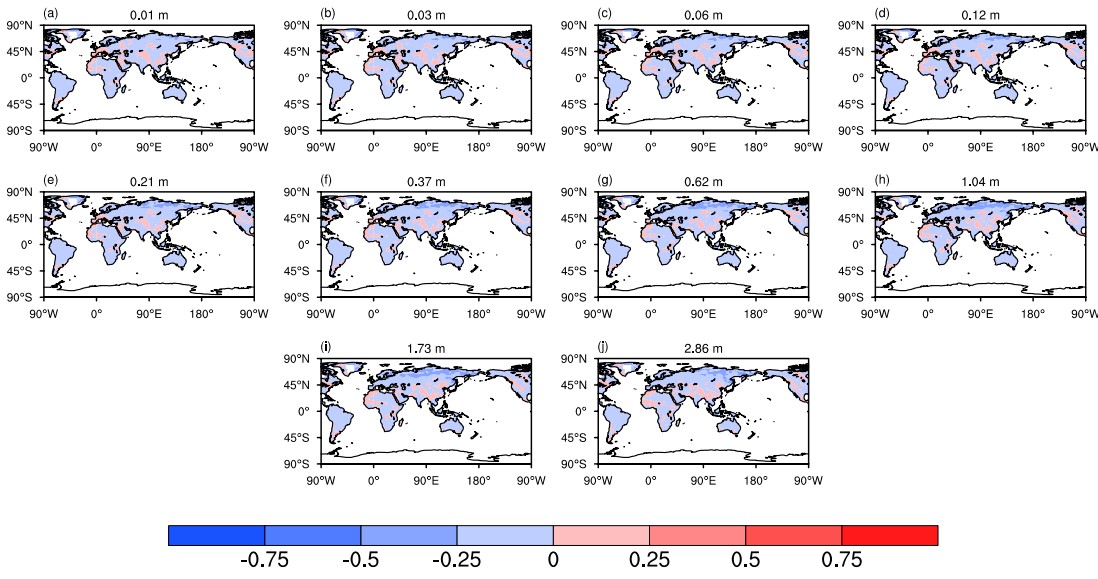


**Figure 5.** Same as in Figure 4, but for soil temperature.

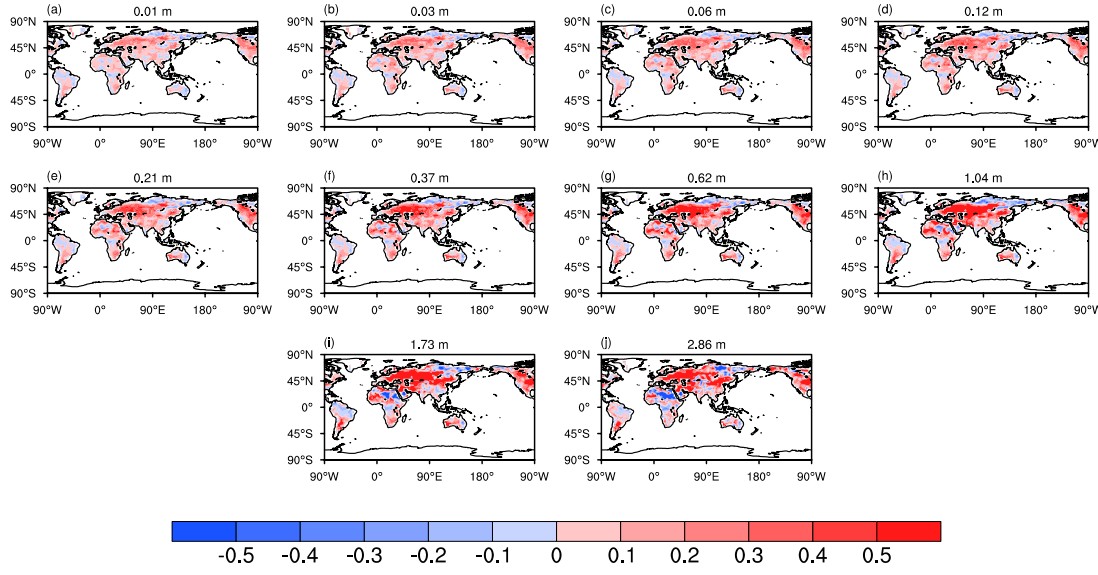


**Figure 6.** Differences between correlations of soil moisture in Assim and CTRL with the GLDAS data

from the surface to deep layers for the period of 1980-2016. The number at the top center denotes the

depth of each soil layer.

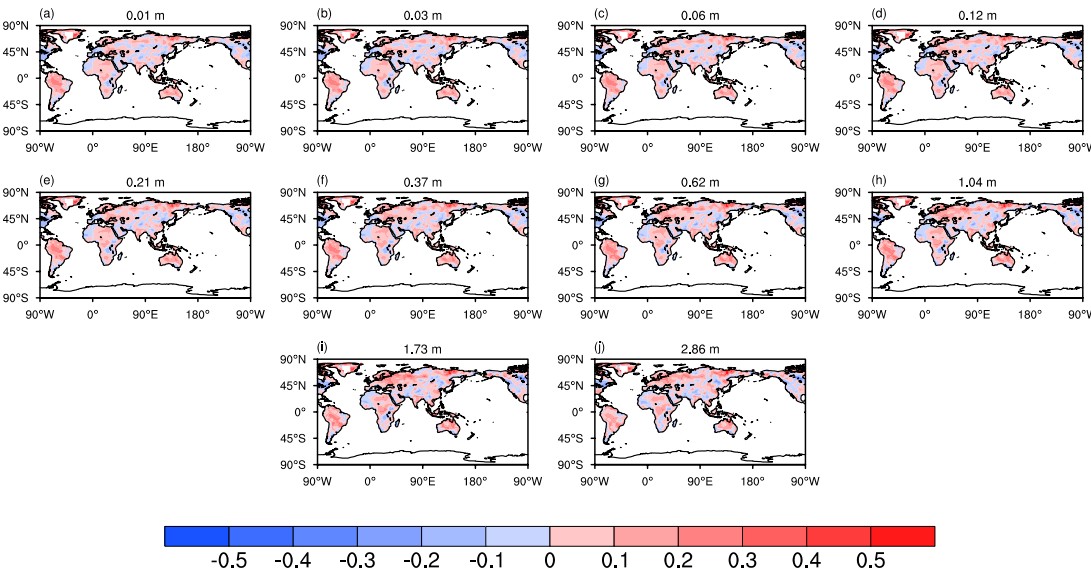


**Figure 7.** Same as in Figure 6, but for soil temperature.

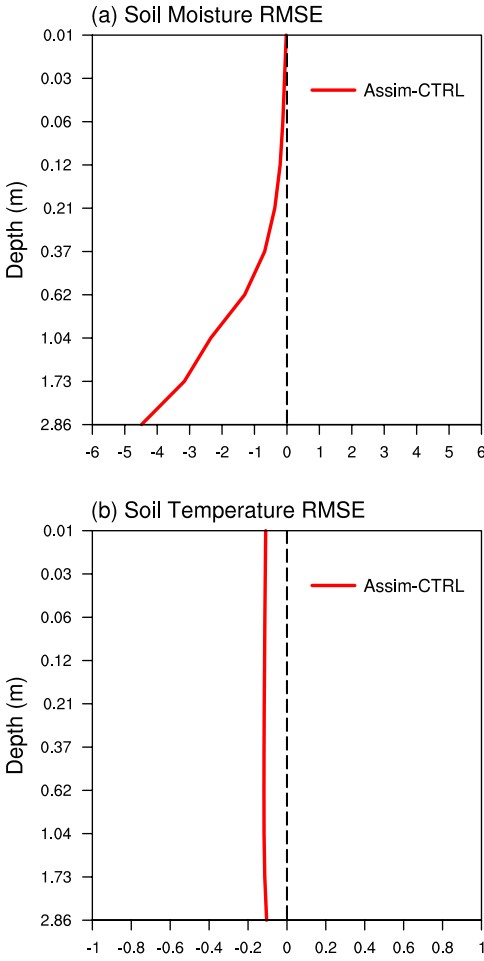


**Figure 8.** Vertical distributions of RMSE differences (Assim minus CTRL) for (a) soil moisture and (b)
soil temperature averaged over the global land during the 1980-2016 period.

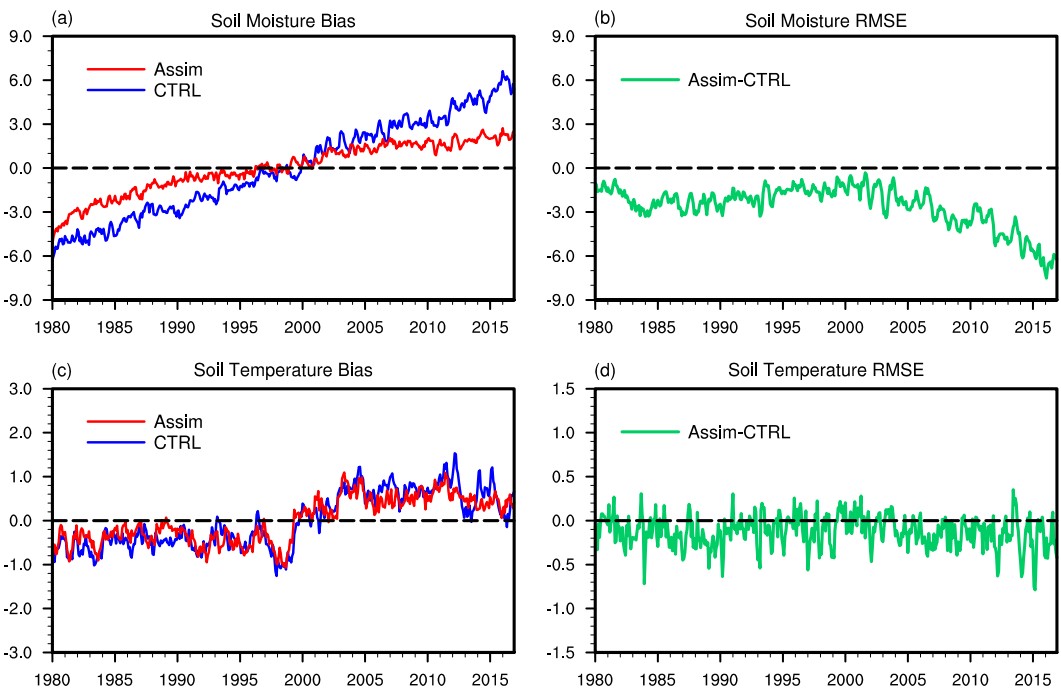


**Figure 9.** Time series of the vertically averaged global mean soil moisture and temperature bias (left) for
Assim (red line) and CTRL (blue line), and RMSE differences (right, green line) between Assim and
CTRL from 1980 to 2016.

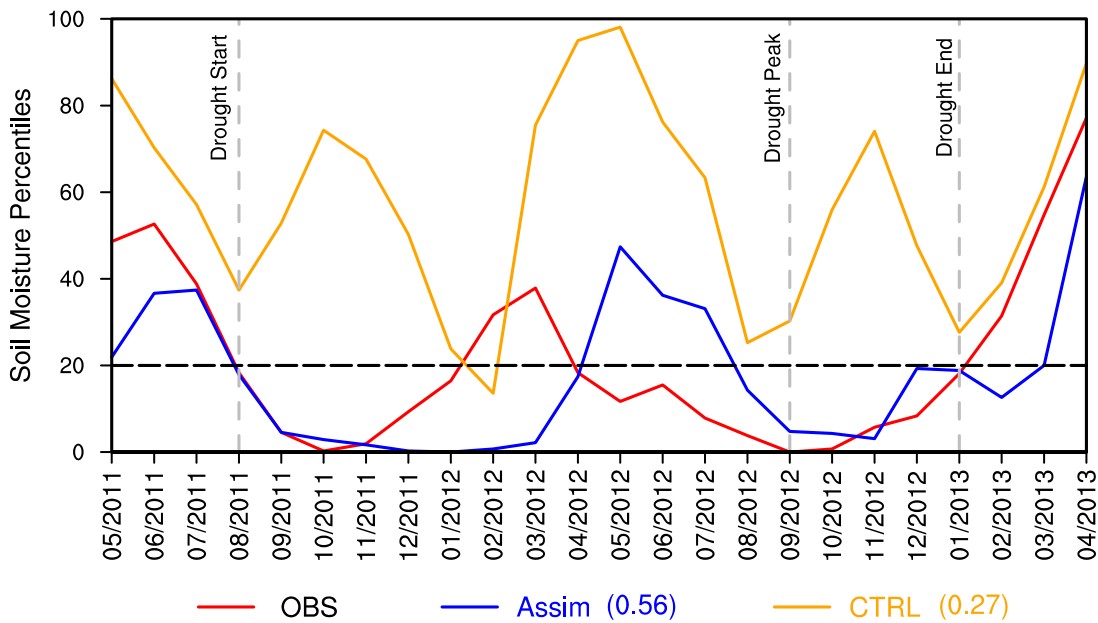


**Figure 10.** Time series of soil moisture percentiles between May 2011 and April 2013 during the 2012

U.S. Midwest drought. Red line: observation, blue line: Assim, orange line: CTRL. The correlation

coefficients of Assim and CTRL with observations are also shown. The three vertical dashed lines mark

the timing of drought start, drought peak and drought end, respectively. The start of the agricultural

drought is defined as the month when soil moisture falls below the 20th percentile. The soil moisture

percentiles are averaged over the U.S. Midwest (36°-50°N, 102°-88°W). The observed soil moisture is

derived from ERA-Interim monthly soil moisture data.

**Appendix A: Supporting Information**

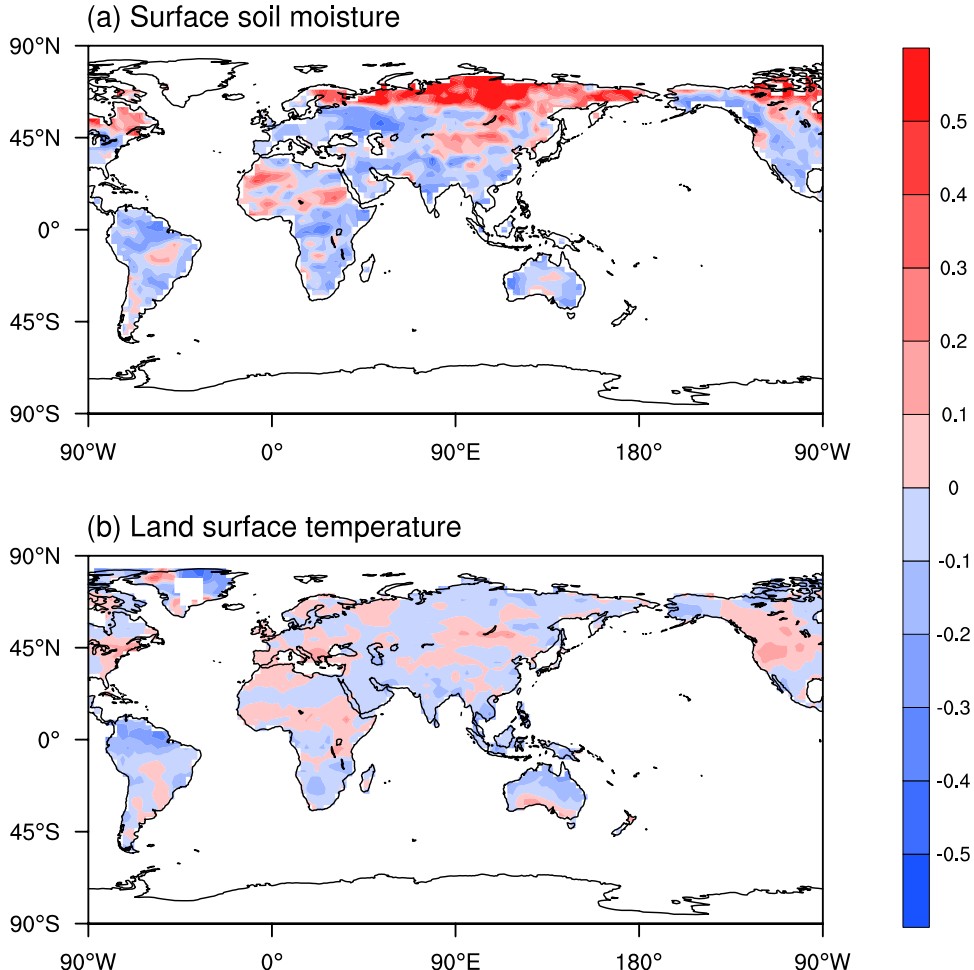


**Figure A1.** Spatial distribution of the AE index for (a) surface soil moisture and (b) land surface
temperature during the 2003-2014 period. The surface soil moisture and land surface temperature are
derived from monthly AMSR and MODIS satellite data, respectively.