# Peer review of "The 4DEnVar-based weakly coupled land data assimilation system for E3SM version 2"

_Geoscientific Model Development, 2023_

## Author Comment (AC1)

We thank Reviewer #1 for the constructive comments and suggestions, which greatly help to improve the quality of our manuscript. We have made revisions and replied to all the comments. Please find the point-by-point responses to the comments. Our responses are shown in "Blue" and the changes in the manuscript are shown in "Red".

**Response to the comments from Reviewer #1**

**Comment#1:**
Abstract: Please specify the reanalysis data you assimilate and improvements of soil moisture and temperature simulations.

**Response:**
Thank you for your suggestion. In response to your comment, we have clarified in the abstract that the assimilated data is derived from the Global Land Data Assimilation System (GLDAS) reanalysis (L18-21). Additionally, we have provided more specific details (L26-29) regarding the improvements resulting from our soil moisture and temperature assimilation. In terms of both soil moisture and temperature, the assimilation experiment outperforms the control simulation with reduced RMSE and higher temporal correlation in many regions, especially in South America, Central Africa, Australia, and large parts of Eurasia.

In light of your feedback, we have incorporated mentions of the specific "GLDAS" reanalysis (L18-21) and more detailed descriptions (L26-29) regarding the improvements in the Abstract.

L18-21: With an initial interest in providing initial conditions for decadal climate predictions, monthly mean anomalies of soil moisture and temperature from the Global Land Data Assimilation System (GLDAS) reanalysis from 1980 to 2016 are assimilated into the land component of E3SMv2 within the coupled modeling framework with a one-month assimilation window.

L26-29: In terms of both soil moisture and temperature, the assimilation experiment outperforms the control simulation with reduced RMSE and higher temporal correlation in many regions, especially in South America, Central Africa, Australia, and large parts of Eurasia.

**Comment#2:**
Sub-section 2.2: GLDAS dataset cannot actually be classified as a "observation dataset" since it is generally based on land surface models. Besides, soil moisture derived from different land surface models are systematically different (e.g., different soil moisture range and long-term mean value) which may introduce additional bias into the coupled data assimilation system. How do you handle this problem?

**Response:**
We agree that GLDAS data are land reanalysis data produced by models. Accordingly, we have revised our manuscript to replace the term "Observational Dataset" with "Land Reanalysis Dataset" on line 151.

We would like to clarify that our employed DRP-4DVar method does not input the full information of the GLDAS data into the initial conditions (ICs) of E3SM but rather, only incorporates part of the GLDAS information by fitting reanalysis data with historical samples produced by the model to form consistent forecast states (Wang et al., 2010). In light of your comment, we have further modified our experiment design to add bias correction before assimilation and conduct the anomaly assimilation for the weakly coupled land data assimilation (WCLDA) systems (L168-171). In our revised manuscript, we have updated all of the figures (Figures 3 to 10) along with their corresponding descriptions to represent the assimilation performance with bias correction.

L168-171: In this study, we conduct the anomaly assimilation for the WCLDA system with bias correction applied to GLDAS data before assimilation. For bias correction, the difference between GLDAS data and its long-term average is calculated as anomalies and then added to the simulated model climatology.

**Comment#3:**
Eq. (5): How to represent the cost function? Please add a string or symbol.

**Response:**
In the revised manuscript, we have added equations for "$J_0$" and "$J_1$" (L303) to represent the observational cost function before and after assimilation respectively in Eq. (8).

L303:
$$\begin{cases} \frac{J_1 - J_0}{J_0} \times 100\% \\ J_0 = \frac{1}{2}(y_{obs} - y_b)^T R^{-1}(y_{obs} - y_b) \\ J_1 = \frac{1}{2}(y_{obs} - y_a)^T R^{-1}(y_{obs} - y_a) \end{cases} \tag{8}$$

where $J_0$ and $J_1$ denote the observational cost function before and after assimilation respectively.

**Comment#4:**
Figure 3: How do you explain the temporal dynamics (maybe some seasonal cycles) of the cost function?

**Response:**
We have also noticed the cyclical behavior in the cost function. It has been noted that the assimilation performance diminishes during the spring maybe related to the "spring barrier" (Mu et al., 2007) and subsequently recovers in the summer. This phenomenon might be attributed to intrinsic model limitations. Further analysis is required to fully elucidate the underlying causes.

**Comment#5:**
Figure 4: The explanations summarized in sub-section 3.2 are inadequately for demonstrating

soil moisture degradation over many regions after the coupled data assimilation. It is suspiciously for me that Figure 4a, i and j show similar degradation spatial patterns while Figure 4b-h perform differently. If these degradations are related to GLDAS data quality, the off-line data assimilation results should be degraded over similar regions. I think more interpretations or experiments are necessary to figure out these issues.

**Response:**

Thank you for your thoughtful comments. In Figure 4, assimilation performance is degraded in the northern part of Russia and northern Africa. This is consistent with the findings in other studies that assimilation updates in northern Russia are limited due to the complexities of accurately representing frozen ground and snow processes in high latitudes (Edwards et al., 2007; Ireson et al., 2013). The surface soil moisture is highly susceptible to atmospheric conditions, subsequently affecting the assimilation performance. Furthermore, some degradations found in the deep layers could be attributed to the substantial influence of various terrestrial factors, such as subsurface runoff and interactions with groundwater, similar to the findings in previous studies (Liu and Mishra, 2017; Zeng and Decker, 2009).

In light of your suggestions, we have incorporated more detailed interpretations (L331-339) into the revised manuscript.

L331-339: However, assimilation performance is degraded in the northern part of Russia and northern Africa. This is consistent with the findings in other studies that assimilation updates in northern Russia are limited due to the complexities of accurately representing frozen ground and snow processes in high latitudes (Edwards et al., 2007; Ireson et al., 2013). As surface soil moisture is highly susceptible to atmospheric conditions, assimilation performance of surface soil moisture is limited by the accuracy of atmospheric forcing. Furthermore, some degradations found in the deep layers could be attributed to the substantial influence of various terrestrial factors, such as subsurface runoff and interactions with groundwater, similar to the findings in previous studies (Liu and Mishra, 2017; Zeng and Decker, 2009).

**Comment#6:**

I suggest adding a discussion section and focusing on the preconditions or theory basis for applying coupled data assimilation. For examples, if the land-atmosphere relationship is poorly represented, the improved land surface states may incorrectly influence the atmospheric process; for humid regions, the evaporative regime is typically energy-limited and the assimilation of soil moisture has very limited benefit while soil temperature may more effective. Vice versa for arid regions…

**Response:**

We agree with your opinion that the influence of the weakly coupled land data assimilation system on atmospheric processes may be limited in some domains due to uncertainties of the model parameterizations, particularly in representing land-atmosphere interactions (Zhou et al., 2023). For instance, in humid regions, where the evaporation process is predominantly regulated by energy, the assimilation of soil moisture tends to manifest a relatively small

influence. In contrast, the assimilation of soil temperature may facilitate notable improvements within these regions. This underscores the importance of the unique characteristics and constraints presented by complicated regional conditions in the application of assimilation processes.

In response to your recommendation, we have incorporated this discussion (L450-457) in the revised manuscript.

L450-457: It is possible that the influence of the WCLDA system on atmospheric processes may be limited in some domains due to uncertainties of the model parameterizations, particularly in representing land-atmosphere interactions (Zhou et al., 2023). For example, in humid regions where the evaporation process is predominantly energy-limited, the assimilation of soil moisture tends to exert limited influence. Instead, the assimilation of soil temperature may yield more substantial improvements. This underscores the importance of the unique characteristics and constraints presented by complicated regional conditions in the application of assimilation processes.

**References:**

Edwards, A. C., Scalenghe, R., and Freppaz, M.: Changes in the seasonal snow cover of alpine regions and its effect on soil processes: a review, Quaternary International, 162, 172–181, https://doi.org/10.1016/j.quaint.2006.10.027, 2007.

Ireson, A. M., Van Der Kamp, G., Ferguson, G., Nachshon, U., and Wheater, H. S.: Hydrogeological processes in seasonally frozen northern latitudes: understanding, gaps and challenges, Hydrogeology Journal, https://doi.org/10.1007/s10040-012-0916-5, 2013.

Liu, D., and Mishra, A. K.: Performance of AMSR_E soil moisture data assimilation in CLM4.5 model for monitoring hydrologic fluxes at global scale, Journal of Hydrometeorology, 547, 67–79, https://doi.org/10.1016/j.jhydrol.2017.01.036, 2017.

Mu, M., Xu, H., and Duan, W.: A kind of initial errors related to "spring predictability barrier" for El Niño events in Zebiak-Cane model, Geophysical Research Letters, 34, L03709, https://doi.org/10.1029/2006GL027412, 2007.

Wang, B., Liu, J., Wang, S., Cheng, W., Liu, J., Liu, C., Xiao, Q., and Kuo, Y. H.: An economical approach to four-dimensional variational data assimilation, Advances in Atmospheric Sciences, 27, 715–727, https://doi.org/10.1007/s00376-009-9122-3, 2010.

Zeng, X., and Decker, M.: Improving the numerical solution of soil moisture–based Richards equation for land models with a deep or shallow water table, Journal of Hydrometeorology, 10, 308–319, https://doi.org/10.1175/2008JHM1011.1, 2009.

Zhou, J., Yang, K., Crow, W.T., Dong, J., Zhao, L., Feng, H., Zou, M., Lu, H., Tang, R. and Jiang, Y.: Potential of remote sensing surface temperature-and evapotranspiration-based land-atmosphere coupling metrics for land surface model calibration, Remote Sensing of Environment, 291, 113557, https://doi.org/10.1016/j.rse.2023.113557, 2023.

---

## Author Comment (AC2)

We thank Reviewer #2 for the constructive comments and suggestions, which greatly help to improve the quality of our manuscript. We have made revisions and replied to all the comments. Please find the point-by-point responses to the comments. Our responses are shown in "Blue" and the changes in the manuscript are shown in "Red".

**Response to the comments from Reviewer #2**

**General Comment:**
This paper presents experiments assimilating soil moisture and soil temperature from the GLDAS modeling system into the E3SM model using 4DEnVar. The use of Hybrid DA method for the land DA is unusual, and is an interesting development that I am curious to see more work on. Unfortunately, the experimental design is badly flawed, and the information presented in the paper is vague, out-dated, very often incorrect, and difficult to follow.

**Response:**
Thank you very much for taking time to review our manuscript and providing us very useful comments. We are sorry for any confusions or misrepresentations that our initial draft might have conveyed. We value your comments and have revised the relevant contents on the experimental design. We have rewritten the introduction section of the manuscript and revised the manuscript carefully according to your comments and suggestions. Please refer to the point-by-point responses in the following.

**Comment#1:**
This work is assimilating model output soil moisture and soil temperature into a different model, with no accounting for the systematic differences between the two models. Data assimilation is not typically applied to assimilate fields from one model into another (unless conducting a synthetic twin experiment to test aspects of the DA, which is not how this is presented). There is also extensive literature discussing the fact that soil moisture cannot be transferred from one model to another without rescaling it, which makes the approach here invalid. For example, see: RD Koster, Z Guo, R Yang, PA Dirmeyer, K Mitchell, MJ Puma, On the nature of soil moisture in land surface models, Journal of Climate 22 (16), 4322-4335. And many references in that paper. Additionally, the assimilated fields are monthly means, which is not the obvious choice, and this is not adequately discussed. To be publishable, the authors would need to assimilate actual observations, not model output, and would need to apply adequate bias correction/rescaling to those observations (particularly to soil moisture - see work by Rolf Reichle, Randy Koster, etc).

**Response:**
Many thanks to your detailed comment on the data assimilation (DA) approach used in our study. We recognize that DA is widely applied to produce high-quality real-time reanalysis data by assimilating actual observations such as satellite data for numerical weather predictions (NWPs). However, in this study, the weakly coupled data assimilation (WCDA) system we built is intended to be used to initialize decadal climate predictions (DCPs) with an earth system model that fully couples various component models including the atmospheric model, land

surface model, oceanic model, sea ice model, and so on. The initialization for DCPs is quite different from that of NWPs, primarily due to the large difference in temporal scales. Almost all initializations for DCPs in CMIP5 and CMIP6 incorporated monthly mean reanalysis data as observations (Table 1). This preference is based on two key considerations. Firstly, for decadal-scale applications, data signals with temporal resolutions shorter than one month could introduce undesirable noise, which can adversely affect DCPs when high temporal resolution data are assimilated into the ICs. This is why almost all initialization approaches for DCPs used in CMIP5 and CMIP6 assimilate monthly mean data. Secondly, the DA approaches utilized in the coupled data assimilation (CDA) for initializations of decadal prediction are generally much simpler than those used in NWPs, due to the complexity of the coupled model. For examples, many initialization systems used in CMIP5 and CMIP6 adopted the nudging method (Table 1). Because of the simpler DA approaches and more complex coupled models, actual observations cannot be assimilated directly into the WCDA system. Furthermore, unlike NWPs where long-term DA cycles aren't necessary, the initialization for DCPs requires DA cycles of at least ten years which makes it very difficult or even impossible to assimilate actual observations due to the very high computational cost.

**Table 1.** Brief summaries of assimilation strategies used in CMIP5 and CMIP6 decadal prediction experiments through assimilation of reanalysis data.

| Model | Assimilation Strategies | Method | References |
|---|---|---|---|
| BCC-CSM1.1 | Ocean: assimilate the SODA reanalysis | Nudging | Xin et al., 2013 |
| CanCM4 | Atmosphere: assimilate the ERA reanalysis | Nudging | Merryfield et al., 2013 |
| CNRM-CM5 | Ocean: assimilate the NEMOVAR reanalysis | Nudging | Voldoire et al., 2014 |
| HadCM3 | Atmosphere: assimilate the ERA-40 reanalysis | Nudging | Smith et al., 2013 |
| FGOALS-g2 | Ocean: assimilate the ds285.3 reanalysis | Nudging | Wang et al., 2013 |
| EC-Earth3 | Ocean: assimilate the ORAS4 reanalysis | Nudging | Bilbao et al., 2021 |
| NorCPM1 | Ocean: assimilate the HadISST reanalysis | EnKF | Bethke et al., 2021 |
| CanE3M5 | Ocean: assimilate the ORAS5 reanalysis | Nudging | Sospedra-Alfonso et al., 2021 |

GLDAS product generates optimal fields of land surface states and fluxes in near-real time because the atmospheric forcing is based on actual observations (Rodell et al., 2004). These reliable and high-resolution global land surface datasets from GLDAS are extensively utilized in weather and climate research (Chen et al., 2021; Zhang et al., 2018). In our search for the most suitable long-term land surface dataset, GLDAS emerged as a top choice. Therefore, we employed the advanced WCDA approach to incorporate the GLDAS monthly mean soil temperature and soil moisture into the fully coupled E3SMv2 model. It is noteworthy that

GLDAS products were also assimilated in another coupled model (FGOALS-g2), showing significant improvements in the interannual prediction skills over East Asia and Europe, as shown in previous studies by Shi et al. (2021, 2022).

Furthermore, it's important to clarify that assimilating the GLDAS data into E3SM does not mean directly transferring soil moisture data from one model to another, because our employed DRP-4DVar method, which is a 4DVar approach, does not input the full information of the GLDAS data in the initial conditions (ICs) of E3SM but instead this approach only incorporates part of the GLDAS information by fitting reanalysis data with historical samples produced by the model to form consistent forecast states (Wang et al., 2010). More detailed descriptions of the DRP-4DVar method can be found in Wang et al. 2010.

We have added a detailed reason for assimilating monthly mean reanalysis (L245-252) in the revised manuscript.

L245-252: In contrast to decadal timescales, data signals with temporal resolutions shorter than one month can potentially introduce undesirable noise, which can adversely affect DCPs when high temporal resolution data are assimilated into the ICs. Moreover, it is very computationally demanding to assimilate complex actual observations in the initialization for DCPs that requires long-term DA cycles. Therefore, similar to most existing initialization approaches for DCPs that assimilate reanalysis data, this study describes the implementation of a data assimilation approach for initializing DCPs by assimilating monthly mean GLDAS data within the one-month assimilation window.

We recognize the current limitations of the weakly coupled land data assimilation (WCLDA) system implemented for E3SMv2. Specifically, our current assimilation system lacks the design of the observation operator, making it challenging to assimilate actual observations (e.g., satellite data) at this stage. The observation operator is important in providing the linkage between the model variables and actual observations that differ in spatial and temporal resolutions. We have noted the design of the observation operator as a direction for future development, and highlighted this limitation in the discussions of the revised manuscript (L446-450).

L446-450: Our current WCLDA system has some limitations such as the lack of an observation operator to integrate actual observations (e.g., satellite and station data). An observation operator is crucial in providing the linkage between the model variables and actual observations, which differ in spatial and temporal resolutions. Hence future exploration will focus on developing observation operators suitable for assimilating various satellite data, such as the AMSR-E and GRACE data.

In light of your comments, we have changed the experiment design to add the bias correction to GLDAS data before assimilation (L168-171), and then conducted the anomaly assimilation through assimilating observed anomalies into the model in the revised manuscript. Due to the modifications of our experimental design, we have revised all of the figures (Figure 3 to 10)

and relevant descriptions to illustrate the assimilation performance with bias correction in our revised manuscript.

L168-171: In this study, we conduct the anomaly assimilation for the WCLDA system with bias correction applied to GLDAS data before assimilation. For bias correction, the difference between GLDAS data and its long-term average is calculated as anomalies and then added to the simulated model climatology.

**Comment#2:**
The background information presented in the introduction demonstrates very little understanding of the standard methods used in land DA and coupled land/atmosphere DA, and presents a picture of modern data assimilation practices that is incorrect. Much of the introduction is also rather vague-with references to coupled and uncoupled systems that are unclear. I recommend completely re-writing the introduction by first identifying the modeling system that you are working with (land /atmosphere?), and introducing examples related to that system. Then, then make sure it is always clear what you are referring to when referencing a coupled model or coupled DA system (with a clear distinction between coupling in the model and coupling in the DA). Of greatest concern, the method is presented as 'coupled data assimilation' but the experiments assimilate land "observations" into only the land component of their model (a coupled land/atmosphere model)-this is not coupled DA! In general, the paper seems confused between coupled modeling, and coupled DA. There's also no mention of weakly or strongly coupled DA, which is very relevant.

**Response:**
We appreciate very much this valuable comment that greatly benefits the improvement of the introduction of our manuscript and the whole paper. According to this comment, we've completely rewritten the introduction. In the revised manuscript, we first elucidate our focus on the initialization of a climate model that fully couples various component models, including atmospheric model, land surface model, ocean model, sea ice model and so on. Subsequently, we delve into both uncoupled data assimilation (DA) and coupled data assimilation (CDA) methodologies. Within the domain of CDA, we draw distinctions between weakly coupled data assimilation (WCDA) and strongly coupled data assimilation (SCDA), providing detailed descriptions for each approach. To address the comment, we have added a description of the fully coupled climate model and the initialization of the climate model (L41-56) in the introduction of the revised manuscript.

L41-56: Much work has been devoted to initializing climate system models for practicable decadal climate predictions (DCPs). These models couple various components, such as models of the atmosphere, land surface, ocean, sea ice, and so on. Due to their much higher complexity, coupled models are often more susceptible to initial conditions (ICs) than their individual model components, underscoring the importance of dedicated data assimilation (DA) (Sakaguchi et al., 2012). The capability of DA methods is essential to incorporate available observations into the components of coupled model and produce the optimal estimate of ICs to improve DCPs. The initialization for DCPs uses uncoupled DA and coupled data assimilation

(CDA) methods. Uncoupled DA performs DA under the framework of an individual component model (e.g., standalone land surface model forced by atmospheric observations or reanalysis data rather than coupled with an atmospheric model), and then the uncoupled DA analyses from different individual components are combined to form the ICs of a coupled model (Zhang et al., 2020). For example, most existing reanalysis data were produced using uncoupled DA approaches, and these reanalysis datasets are then directly used to initialize DCPs in some studies (Du et al., 2012; Bellucci et al., 2013). However, such uncoupled DA often exhibits poor consistency among the ICs of different component models, and eventually produces low prediction skills (Balmaseda et al., 2009; Boer et al., 2016; Ardilouze et al., 2017).

We have also revised our Introduction (L57-80) to include descriptions of both weakly coupled data assimilation (WCDA) and strongly coupled data assimilation (SCDA). We distinguish between WCDA and SCDA by highlighting the characteristics of coupling in the model and coupled DA.

L57-80: To obtain balanced multi-component ICs in coupled models, recent studies focus on the development of CDA methods under the coupled modeling framework (Penny and Hamill, 2017; He et al., 2020a). The purpose of CDA is to produce balanced and coherent ICs for all components within the climate system by incorporating observational information from one or more components in the coupled model, providing great potential for improving seamless climate predictions (Dee et al., 2014). Some studies underscore the superior advantages of CDA over traditional uncoupled DA methods (Lea et al., 2015; Zhang et al., 2005). CDA methods are categorized into two main types: weakly coupled data assimilation (WCDA) and strongly coupled data assimilation (SCDA). WCDA assimilates the observations or existing reanalysis into the respective component of the coupled model and then transfers the observational information to the other components through the coupled model integration (He et al., 2020b; Zhang et al., 2020). Considering that sequential DA encompasses both the analysis and the forecast steps, WCDA allows no direct influence of observations from a single component to other components in the analysis step as the cross-component background error covariances are not used, but coupling in the forecast step allows interactions across different components during the model integration (Browne et al., 2019) and propagates the observational information to other components. In contrast, SCDA utilizes cross-component background error covariances to directly assimilate the observational information from one component into all components, treating the entire Earth system model as one unified system (Penny et al., 2019). Furthermore, similar to WCDA, SCDA also allows coupling in the forecast step to propagate the observations from one component to the other components (Yoshida and Kalnay, 2018). Several studies indicate that SCDA typically exhibits more pronounced improvements in assimilation performance relative to WCDA (Smith et al., 2015; Sluka et al., 2016). However, the application of SCDA poses substantial technical challenges, particularly in the establishment of effective cross-component background error covariances. Consequently, the majority of contemporary CDA systems still utilize the WCDA framework.

According to the aforesaid introduction to uncoupled DA and CDA, their main difference is the use of forecast model at the forecast step. If the forecast model is the coupled model, the

DA is categorized as CDA. Otherwise, if the forecast model is an individual component model, the DA is referred to as uncoupled DA. Additionally, it's important to note that CDA does not necessarily require assimilating observations from multiple components. Even if CDA assimilates the observations just from a single component, it can still enhance the ICs of all components within the coupled model.

In this study, the WCDA system was built and used to produce ICs for all components of E3SM by incorporating GLDAS reanalysis. During the analysis step, our WCDA system only assimilates land reanalysis data into the land component of an earth system model. However, it's crucial to highlight that in the forecast step, the entire E3SM climate model rather than the land surface model is used as the forecast model to forecast the IC backgrounds of all components for the next assimilation window and the land reanalysis information can propagate to the other components (e.g., atmosphere and ocean) dynamically through the coupled integration of E3SM during the one-month forecast. In general, when the coupled model is used in the forecast step while the optimal assimilation analysis is updated separately for the respective component, the assimilation approach is identified as WCDA (Penny et al., 2019; Zhang et al., 2020). Thus, the assimilation approach in this study is referred to as a WCDA system.

We have added a detailed description (L221-228, L268-278 and L288-294) of the implementation of weakly coupled assimilation system and updated the name of our data assimilation system from CDA to WCDA throughout the manuscript. This modification is intended to provide a clearer depiction of the WCDA system in this manuscript.

L221-228: In the analysis step, only the land state variables are updated to the optimal analysis ($x_a^{lnd}$). Subsequently, we proceed with a one-month freely coupled integration of the E3SMv2 model during the forecast step. This integration is initialized from the optimal land ICs ($x_a^{lnd}$) along with the background fields as the ICs of other components (e.g., atmosphere and ocean). Throughout this one-month free integration, the interactions among the model components indirectly enhance the background states of these components (e.g., atmosphere and ocean) for the next assimilation window due to the more realistic land state variables. Moreover, this coupled integration also contributes to the balance between the ICs of different components.

L268-278: The incorporation of GLDAS data into the E3SMv2 model consists of the analysis step and the forecast step. In the analysis step, the differences between monthly mean GLDAS data and model outputs are calculated and utilized to produce the optimal assimilation analysis at the beginning of a one-month assimilation window. Subsequently, in the forecast step, this optimal assimilation analysis is used as the land ICs combined with the background ICs for other components to conduct one-month forecast using the E3SMv2 model. This forecast generates the backgrounds of all model components for the next assimilation window. As a result, the forecasted backgrounds for all components are influenced by the land reanalysis information incorporated into the ICs of the land component. In general, when the coupled model is used in the forecast step while the optimal assimilation analysis is updated separately

for the respective component, the assimilation approach is identified as WCDA (Penny et al., 2019; Zhang et al., 2020).

L288-294: To assimilate the monthly mean GLDAS product, fully coupled integration by the E3SMv2 model is performed twice within each one-month assimilation window: first to generate the observational innovation by computing the differences between the GLDAS data and model outputs for analysis, and second to forecast the backgrounds of all components for the next assimilation window. When the fully coupled model is executed for the second one-month run, the land reanalysis information is transferred to the other components through multi-component interactions.

**Comment#3:**
Paragraph starting L37. I had a lot of trouble following the argument in this paragraph, largely because the terms used have not been defined (and I suspect are not being applied in their standard usage). The distinction between coupling in the model and in the DA is unclear, and I really don't know what the "uncoupled" option is. Perhaps using an explanatory example might help.

**Response:**
We are sorry for any confusion caused. We understand your concerns about the unclear distinction between coupling in the model and the DA. In the revised manuscript, we have rewritten this paragraph (L41-56 in the revised manuscript) to clearly differentiate between uncoupled and coupled data assimilation. Uncoupled DA implies that DA is conducted using a standalone component model (e.g., a standalone land surface model forced by atmospheric reanalysis data or observations rather than coupled with an atmospheric model) as the forecast model to produce the IC background, and then the uncoupled DA analyses from different components are combined to form the ICs of the coupled model (Zhang et al., 2020). For example, most existing reanalysis data were produced by uncoupled DA, and some decadal predictions were initialized directly using reanalysis data for different components (Du et al., 2012; Bellucci et al., 2013).

We have introduced a new paragraph (L57-80 in the revised manuscript) dedicated to elucidating the concept of coupled data assimilation (CDA), as explained in our response to the last comment.

**Comment#4:**
L38: I'm not sure what is meant by "uncoupled initialization" here or "uncoupled data assimilation". I suspect you mean weakly coupled data assimilation (as in separate DA systems for each model component, each assimilating obs from that component) - which is still coupled DA.

**Response:**
We have revised the relevant descriptions (L48-54) to provide a clearer definition on "uncoupled initialization". In this context, "uncoupled initialization" is not the same as WCDA.

Instead, in "uncoupled initialization" reanalysis data are directly used to initialize the individual components of the coupled models, and this approach is defined as uncoupled DA.

L48-54: Uncoupled DA performs DA under the framework of an individual component model (e.g., standalone land surface model forced by atmospheric observations or reanalysis data rather than coupled with an atmospheric model), and then the uncoupled DA analyses from different individual components are combined to form the ICs of a coupled model (Zhang et al., 2020). For example, most existing reanalysis data were produced using uncoupled DA approaches, and these reanalysis datasets are then directly used to initialize DCPs in some studies (Du et al., 2012; Bellucci et al., 2013).

We agree with your opinion about the definition of weakly coupled data assimilation. In this study, soil moisture and temperature data from GLDAS are assimilated into the land component of the climate model during the analysis step. Subsequently, during the forecast step, the entire E3SM climate model is used as the forecast model to forecast the IC backgrounds of all components for the next assimilation window and the observed land information is transferred to the other components (e.g., atmosphere and ocean) through the coupled model integration. Thus, our DA process under the coupled modeling framework is referred as the WCDA system.

**Comment#5:**
L39: what is a "stand-alone" model state?

**Response:**
"Stand-alone" model refers to the individual component models, such as the atmospheric or land surface models, which operate independently without interactions with other components. For example, a standalone land surface model is forced by atmospheric observations or reanalysis data but not coupled with an atmospheric model.

**Comment#6:**
L40: I assume the Prodhomme paper refers to a hydrological system, so it's a bit misleading to say "some modeling centers …" and then refer to a single product that is not their main product.

**Response:**
Thank you for bringing this to our attention. The phrase "some modeling centers" has been revised to "some studies" for clarity. Furthermore, we have decided to remove the citation of the Prodhomme paper and replaced it with references to other relevant articles. This sentence (L51-54) is revised as "For example, most existing reanalysis data were produced using uncoupled DA approaches, and these reanalysis datasets are then directly used to initialize DCPs in some studies (Du et al., 2012; Bellucci et al., 2013)."

In response to this comment, we have made adjustments to this sentence (L51-54) in the revised manuscript.

L51-54: For example, most existing reanalysis data were produced using uncoupled DA

approaches, and these reanalysis datasets are then directly used to initialize DCPs in some studies (Du et al., 2012; Bellucci et al., 2013).

**Comment#7:**
L41: again, I don't know what you mean by "uncoupled methods". If you're applying DA to multiple model components in a coupled model, there is some coupling introduced via the model forecasts - at a minimum, this is weakly coupled DA.

**Response:**
The "uncoupled methods" here refer to the direct use of reanalysis data as ICs for the coupled model. We have explained what "uncoupled DA" implies. Please refer to the responses to Comments #3 and #4 for the details.

According to your comments, we have revised the manuscript (L48-54) to explicitly define the "uncoupled DA", and we have also added detailed descriptions (L113-118) to emphasize that the data assimilation system employed in our study is a weakly coupled DA system. This will help delineate the differences between the uncoupled methods we initially mentioned and the weakly coupled approach we utilized.

L48-54: Uncoupled DA performs DA under the framework of an individual component model (e.g., standalone land surface model forced by atmospheric observations or reanalysis data rather than coupled with an atmospheric model), and then the uncoupled DA analyses from different individual components are combined to form the ICs of a coupled model (Zhang et al., 2020). For example, most existing reanalysis data were produced using uncoupled DA approaches, and these reanalysis datasets are then directly used to initialize DCPs in some studies (Du et al., 2012; Bellucci et al., 2013).

L113-118: In this WCLDA system, monthly mean anomalies of soil moisture and temperature from a global land reanalysis product are assimilated into the land component of a coupled climate model in the analysis step, and subsequently during the forecast step, the land reanalysis information incorporated into the ICs of the land component is propagated to the other components (e.g., atmosphere and ocean) through the fully coupled model integration and influences the ICs of all components for the next assimilation window.

**Comment#8:**
L44 "each coupled model individually". Do you mean each component of the coupled model?

**Response:**
"Each coupled model individually" refers to each climate system model, rather than each component of the coupled model. What we intended to express was that distinct DA systems could be constructed for different climate models to obtain high-quality and well-balanced initial conditions. To avoid further ambiguity, we have removed this sentence in the revised manuscript.

**Comment#9:**

L50: Again, without concrete examples it is hard to follow the argument in the paragraph. I am unclear what the actual application being discussed is, since the text just refers to "modeling centers" or "models". The references come largely from atmospheric DA, with some coupled atmosphere / other component examples. If this is the case, then the information presented is this paragraph about the usage of different methods is largely incorrect and very outdated.

**Response:**

According to your comments, we have made consistent modifications throughout the manuscript, replacing "modeling centers" with more appropriate terms like "some studies" and "previous studies". This ensures a clearer depiction of the source of our references and the focus of our work.

Furthermore, to improve clarity, we have started this paragraph by emphasizing the use of coupled data assimilation methods to initialize decadal climate predictions (DCPs) (L81-82): "Recent research efforts have started to implement the CDA system to initialize DCPs, using a diverse range of DA techniques from simple to complex." Based on our careful review of literature, most of DCPs do not assimilate land observations or reanalysis data.

We have rewritten this paragraph (L81-108) to introduce the development of CDA methods by providing more recent and relevant literature.

L81-108: Recent research efforts have started to implement the CDA system to initialize DCPs, using a diverse range of DA techniques from simple to complex. The simplest method is nudging which adjusts the model states towards the observations or existing reanalysis (Hoke and Anthes, 1976; Zhang et al., 2020). Although the nudging method is time-saving and easy to implement, its application in CDA is restricted primarily due to the limited types of observations and the required interpolation of observations at every time step of model integration (He et al., 2017). Previous studies have developed advanced CDA systems using variational and filtering approaches, such as the three-dimensional variational data assimilation (3DVar) (Laloyaux et al., 2016; Yao et al., 2021), and ensemble-based techniques like the ensemble Kalman filter (EnKF) (Zhang et al., 2007). The former generally utilizes the stationary background error covariance and assimilates observations sequentially (Lin et al., 2017). In contrast, the latter uses the flow-dependent forecast error covariance and recursively integrates observations into the model (Lei and Hacker, 2015). Several studies also show encouraging progress in constructing CDA systems using four-dimensional variational data assimilation (4DVar) method (Smith et al., 2015; Fowler and Lawless, 2016). The objective of 4DVar is to optimize four-dimensional model states and provide a compatible temporal trajectory that matches observational records across each assimilation window (Mochizuki et al., 2016). The 4DVar method is an advanced assimilation technique that exhibits superiority over other assimilation techniques like nudging and 3DVar in multiple aspects. Initial shocks that influence prediction skills can be significantly minimized by the 4DVar approach due to the dynamical consistency between the model and ICs (Sugiura et al., 2008). However, it is difficult to apply the 4DVar method for CDA systems in the fully coupled model because of

the challenge in adjoint integration of the coupled model and its high computational cost in the analysis step. Finally, to capitalize on the strengths of both ensemble and variational techniques, recent studies focus on developing new hybrid data assimilation methods (Wang et al., 2010; Buehner et al., 2018). The hybrid approach utilizes an ensemble forecast to generate flow-dependent forecast error covariances and presents a way to perform 4DVar optimization without the need for tangent linear and adjoint models (Lorenc et al., 2015). However, most studies on CDA have focused on assimilating observations or reanalysis data of ocean, atmosphere and even sea ice. There have been relatively few instances of CDA studies assimilating land observations or reanalysis data.

**Comment#10:**
L52: Here you are talking about the method used to add the analysis increment to the model states, which is a secondary detail to how the increment is calculated - the discussion is very misleading. For example, the Bloom paper you cite is from NASA GMAO. They currently use 4DEnVar to calculate the increment, and IAU to add it. You may also be mixing up the "nudging" DA method with nudging methods to add increments. If so, nudging is also a very old DA technique, and has not been standard use for decades.

**Response:**
In light of this comment, we have revised our overview of the nudging methods (L82-86). Furthermore, we have removed the citation to the Bloom paper and have replaced it with more recent and relevant references. Additionally, we have highlighted the limitations and potential drawbacks associated with nudging to offer a comprehensive perspective.

L82-86: The simplest method is nudging which adjusts the model states towards the observations or existing reanalysis (Hoke and Anthes, 1976; Zhang et al., 2020). Although the nudging method is time-saving and easy to implement, its application in CDA is restricted primarily due to the limited types of observations and the required interpolation of observations at every time step of model integration (He et al., 2017).

**Comment#11:**
L54: IAU (and nudging schemes I know of ) do not "recover the observations". They move the model towards the observations, by some amount determined by the respective observation and background errors.

**Response:**
We recognize that this phrasing "recover the observations" is not accurate in describing the nudging and IAU methods. We have removed this phrasing from the revised manuscript to ensure clarity and accuracy.

**Comment#12:**
L59: The information presented here is very outdated. Major NWP / reanalysis all use hybrid DA methods now - 3Dvar was two generations of DA schemes ago. I'm not familiar with the

Yao paper, but the Lin paper was conducted at a university, not a "modeling center". Likewise the Santonello paper - that's a research paper, not linked to an operational DA system.

**Response:**

We have replaced "some modeling centers" with "previous studies". This sentence (L86-87) is revised as "Previous studies have developed advanced CDA systems using variational and filtering approaches". Although we acknowledge that major NWPs now utilize hybrid DA methods, our focus in this paragraph is on the research progress of coupled data assimilation systems in coupled models. The development of coupled data assimilation in the coupled models is still at its early stages (Zhang et al., 2020) compared to DA used in NWPs due to the complexity of the coupled models. Additionally, we have removed the citation to the Santonello paper and have chosen to cite other relevant research articles instead.

In light of this comment, we have revised the sentence (L86-89) in the revised manuscript.

L86-89: Previous studies have developed advanced CDA systems using variational and filtering approaches, such as the three-dimensional variational data assimilation (3DVar) (Laloyaux et al., 2016; Yao et al., 2021), and ensemble-based techniques like the ensemble Kalman filter (EnKF) (Zhang et al., 2007).

**Comment#13:**

L71: This sentence is wrong (or at least, extremely misleading). If we're talking about Earth system modeling - so NWP, reanalysis, etc (which the references imply is what we're talking about) then most centers did use 4DVar, but have now moved on to more sophisticated hybrid methods.

**Response:**

We have removed this sentence "few modeling centers utilize 4DVar-based initialization methods". To better convey our point, we have revised this sentence (L99-100) as "However, it is difficult to apply the 4DVar method for CDA systems in the fully coupled model" to emphasize the challenges of applying the 4DVar method.

In our revised manuscript, we have revised this sentence (L99-101) accordingly.

L99-101: However, it is difficult to apply the 4DVar method for CDA systems in the fully coupled model because of the challenge in adjoint integration of the coupled model and its high computational cost in the analysis step.

**Comment#14:**

L121: how many soil layers?

**Response:**

Totally ten soil layers. We have revised this sentence (L152-153) that monthly mean soil moisture and soil temperature data in a total of ten soil layers are produced by the Global Land Data Assimilation System (GLDAS; Rodell et al., 2004).

L152-153: Monthly mean soil moisture and soil temperature data in a total of ten soil layers are produced by the Global Land Data Assimilation System (GLDAS; Rodell et al., 2004).

**Comment#15:**
L123:   GLDAS does not produce observations! These are modeled output.

**Response:**
We agree that GLDAS data are land reanalysis data from model outputs. In response to your feedback, we have revised the term "Observational Dataset" to "Land Reanalysis Dataset" (L151).

L151: 2.2 Land Reanalysis Dataset

**Comment#16:**
There is not enough information here on the 4DEnVar / DRP-4Dvar technique for the reader to understand how it works. Also, how is the ensemble created? How do you ensure the ensemble has reasonable spread near the land? How do you estimate the B matrix?

**Response:**
The DRP-4DVar method has been extensively introduced in Wang et al. (2010), Shi et al. (2021, 2022) and He et al. (2017, 2020) in the context of its application in another climate system model FGOALS-g2. Therefore, we did not provide a detailed introduction to the DRP-4DVar method. In the revised manuscript, we have rewritten Section 2.3 "Data Assimilation Scheme" to provide a comprehensive and clear introduction to the DRP-4DVar method (L178-228), especially to the process of ensemble creation and the estimation of the background error covariance matrix $B$.

L178-228: DRP-4DVar is an economical approach that minimizes the cost function of the standard 4DVar by using the ensemble technique instead of the adjoint technique (Wang et al., 2010). The background error covariance matrix $B$ is estimated using the pure ensemble covariance. The ensemble members originate from historical or ensemble forecasts. Considering the high computational cost of ensemble forecasts for the coupled model in our study, we utilize outputs from the pre-industrial control (PI-CTRL) experiment of E3SMv2 to generate ensemble members. The instantaneous state at the beginning of each month and the corresponding monthly mean state of this month from the 100-year balanced PI-CTRL simulation are used as the samples of initial condition ($x_i$) and forecast samples ($y_i$). The corresponding perturbation samples are calculated as $x_i' = x_i - \bar{x}$ and $y_i' = y_i - \bar{y}$, where $\bar{x}$ and $\bar{y}$ are the 100-year average values of $x_i$ and $y_i$ at the same month, respectively. Then, $m$ pairs of perturbation samples $(x_1', x_2', x_3', \cdots, x_m')$ and $(y_1', y_2', y_3', \cdots, y_m')$ are selected at each DA analysis step according to the correlations between $y_i'$ and the observational

innovation $y'_{obs} = y_{obs} - y_b$ and the independence between $y'$ samples. In this study, $m = 30$. Then the estimation of the background error covariance matrix $B$ is represented by the formula in Eq. (1), utilizing the selected $x'$ samples. To remove the spurious remote correlations in the $B$ matrix, the localization approach is applied to optimize the assimilation performance (Wang et al., 2018).

$$
\begin{cases}
B = bb^T \\
b = \dfrac{1}{\sqrt{m-1}} \times (x'_1 - \bar{x}', x'_2 - \bar{x}', x'_3 - \bar{x}', \cdots, x'_m - \bar{x}') \\
\bar{x}' = \dfrac{1}{m}(x'_1 + x'_2 + x'_3 + \cdots + x'_m)
\end{cases} \tag{1}
$$

According to Wang et al. (2010), DRP-4DVar produces the analysis increment ($x'_a$) by minimizing the 4DVar cost function in the incremental form (Courtier et al., 1994):

$$
\begin{cases}
J(x'_a) = \min\limits_{x'} J(x') \\
J(x') = \frac{1}{2}(x')^T B^{-1} x' + \frac{1}{2}(\tilde{y}' - \tilde{y}'_{obs})^T (\tilde{y}' - \tilde{y}'_{obs})
\end{cases} \tag{2}
$$

Here $x' = x - x_b$ represents the increment of model variables relative to the background; $\tilde{y}'_{obs} = r^{-1} y'_{obs} = r^{-1}(y_{obs} - y_b)$ denotes the weighted observational innovation for monthly mean anomalies of soil moisture and temperature, and $R = rr^T$ is the observational error covariance matrix that is usually diagonal; $\tilde{y}' = r^{-1} y' = r^{-1}(y - y_b)$ is the weighted projection of the increment ($x'$) onto the observation space; the superscript $T$ represents the transpose.

To simplify the calculation of the minimization, the increment of model state variables $x'$ and the corresponding weighted observation increment $\tilde{y}'$ are projected onto the dimension-reduced sample space through the following projection transformations:

$$
\begin{cases}
x' = P_x \alpha \\
\tilde{y}' = P_y \alpha
\end{cases} \tag{3}
$$

where $\alpha$ is the $m$-dimension column vector containing the weight coefficients $(\alpha_1, \alpha_2, \alpha_3, \cdots, \alpha_m)$; $P_x$ and $P_y$ denote the projection matrices that incorporate the initial condition perturbations and the corresponding monthly mean samples as follows:

$$
\begin{cases}
P_x = (x'_1, x'_2, x'_3, \cdots, x'_m) \\
P_y = (\tilde{y}'_1, \tilde{y}'_2, \tilde{y}'_3, \cdots, \tilde{y}'_m)
\end{cases} \tag{4}
$$

where $\tilde{y}'_i = r^{-1} y'_i$ ($i = 1, 2, \cdots, m$). Then the original 4DVar cost function defined in Eq. (2) is transformed into the following new cost function and the analysis can be computed in the sample space by minimizing this new cost function:

$$
\begin{cases}
\tilde{J}(\alpha_a) = \min\limits_{\alpha} \tilde{J}(\alpha) \\
\tilde{J}(\alpha) = \frac{1}{2}\alpha^T B_\alpha^{-1} \alpha + \frac{1}{2}(P_y \alpha - \tilde{y}'_{obs})^T (P_y \alpha - \tilde{y}'_{obs}) \\
x_a = x_b + x'_a = x_b + P_x \alpha_a
\end{cases} \tag{5}
$$

The solution to this minimization problem is formulated as:

$$
\alpha_a = (B_\alpha^{-1} + P_y^T P_y)^{-1} P_y^T \tilde{y}'_{obs} \tag{6}
$$

In this study, the DRP-4DVar-based WCLDA system is used to incorporate the land reanalysis data only. The optimal analysis for the land state variables ($x_a^{lnd}$) is obtained by adding the

analysis increment $(x'^{lnd}_a)$ to the background of land ICs $(x^{lnd}_b)$, as expressed in Eq. (7):

$$x^{lnd}_a = x^{lnd}_b + x'^{lnd}_a = x^{lnd}_b + P_x\left(B_\alpha^{-1} + P_y^T P_y\right)^{-1} P_y^T \tilde{y}'_{obs} \qquad (7)$$

In the analysis step, only the land state variables are updated to the optimal analysis $(x^{lnd}_a)$. Subsequently, we proceed with a one-month freely coupled integration of the E3SMv2 model during the forecast step. This integration is initialized from the optimal land ICs $(x^{lnd}_a)$ along with the background fields as the ICs of other components (e.g., atmosphere and ocean). Throughout this one-month free integration, the interactions among the model components indirectly enhance the background states of these components (e.g., atmosphere and ocean) for the next assimilation window due to the more realistic land state variables. Moreover, this coupled integration also contributes to the balance between the ICs of different components.

**Comment#17:**
L152: This paragraph implies that there is no atmospheric DA in these experiments? In which case this is not coupled DA. It is land data assimilation into a coupled model.

**Response:**
This study did not include atmospheric DA because our focus is on investigating the role of land component in initialization for climate predictions, with an initial interest in decadal climate predictions, as very few CDA studies incorporated land observations or reanalysis data.

As we mentioned in our response to Comment#2, our data assimilation system is a WCDA system. The CDA system can provides ICs for all components of the coupled model no matter if the assimilated observations (or reanalysis data) are from one or more components. The ICs of all components are influenced by the observations directly or indirectly through the fully coupled model integration. Please refer to our response to Comment #2 for more details.

In light of this comment, we have incorporated additional clarifications (L221-228) to reflect the influence of land DA on other components (e.g., atmosphere and ocean) under the coupled modeling framework.

L221-228: In the analysis step, only the land state variables are updated to the optimal analysis $(x^{lnd}_a)$. Subsequently, we proceed with a one-month freely coupled integration of the E3SMv2 model during the forecast step. This integration is initialized from the optimal land ICs $(x^{lnd}_a)$ along with the background fields as the ICs of other components (e.g., atmosphere and ocean). Throughout this one-month free integration, the interactions among the model components indirectly enhance the background states of these components (e.g., atmosphere and ocean) for the next assimilation window due to the more realistic land state variables. Moreover, this coupled integration also contributes to the balance between the ICs of different components.

**Comment#18:**
L157 it's not clear what "freely coupled" means. Which components are coupled? Likewise "externally forced", which components are externally forcing used for?

**Response:**

The term "freely coupled" here refers to the mode of interaction among the various components of our Earth system model, namely the atmosphere, land, river, ocean, and sea ice. The term "freely coupled" implies that the model components interact dynamically without any restraints. The term "externally forced" refers to "forced by solar radiation, greenhouse gases, aerosols and so on". These external forcings mainly act on the atmospheric component and then influence other components (e.g., land surface, ocean, and sea ice) through their coupling with the atmosphere.

In response to this comment, we have revised our manuscript (L236-239) to clearly state that external forcings are used to drive the fully coupled climate system model, namely the atmosphere, land, river, ocean, and sea ice.

L236-239: In the freely coupled simulation, the various components of the Earth system model, namely the atmosphere, land, river, ocean, and sea ice, interact dynamically without any restraints. The observed external forcing mainly acts on the atmospheric component and then influences other components (e.g., land surface, ocean, and sea ice) through their coupling with the atmosphere.

**Comment#19:**

L187: Some discussion here of how you updated the model states from monthly means would have been useful, as this is not straight forward. There has been a lot of work done on this within the context of assimilation GRACE terrestrial water storage. Also, assimilating monthly means to update instantaneous states is not the obvious way to do it - given that you're assimilating model output, you had the option of assimilation instantaneous output.

**Response:**

To assimilate the monthly mean GLDAS product, fully coupled integration by the E3SMv2 model is performed twice within each one-month assimilation window: first to generate the observational innovation by computing the differences between the GLDAS data and model outputs for analysis, and second to forecast the backgrounds of all components for the next assimilation window. When the fully coupled model is executed for the second one-month run, the land reanalysis information is transferred to the other components through multi-component interactions. Similarly, to assimilate the monthly GRACE-based TWS observations, previous studies employed the "two-step" scheme in which the land model integration is performed twice within the same month (Houborg et al., 2012; Girotto et al., 2016). The primary reason for assimilating monthly averages rather than updating instantaneous states is that observational information on timescales shorter than one month can potentially introduce undesirable noise, adversely affecting DCPs upon assimilation into the ICs.

In light of this comment, we have included more detailed discussions (L288-297) on how we assimilate monthly mean GLDAS data and previous work about assimilating monthly GRACE data in our revised manuscript.

L288-297: To assimilate the monthly mean GLDAS product, fully coupled integration by the E3SMv2 model is performed twice within each one-month assimilation window: first to generate the observational innovation by computing the differences between the GLDAS data and model outputs for analysis, and second to forecast the backgrounds of all components for the next assimilation window. When the fully coupled model is executed for the second one-month run, the land reanalysis information is transferred to the other components through multi-component interactions. This approach is similar to previous studies that employed the "two-step" scheme in which the land model integration is performed twice within the same month to assimilate the monthly GRACE-based TWS observations (Houborg et al., 2012; Girotto et al., 2016).

---

## Author Comment (AC3)

We thank Reviewer #3 for the constructive comments and suggestions, which greatly help to improve the quality of our manuscript. We have made revisions and replied to all comments. Please find the point-by-point responses to the comments. Our responses are shown in "Blue" and the changes in the manuscript are shown in "Red".

**Response to the comments from Reviewer #3**

**General Comment:**
This manuscript presents the implementation of a 4DEnVAR method in the E3SMv2. The authors assimilate monthly mean soil moisture and temperature from a land re-analysis product and evaluate the performance of the new data assimilation system vs a control experiment (no assimilation). I find the approach of 4DEnVAR for land data assimilation very interesting. However, there are several shortcomings of the paper that need to be addressed before it is ready to be published in GMD.

**Response:**
We would like to express our sincere gratitude for your time and effort in reviewing our manuscript. We truly appreciate your constructive comments and suggestions, which have significantly contributed to enhancing the quality of our work. We have carefully addressed each comment, as outlined below, and have made the necessary revisions to our manuscript.

**Comment#1:**
The authors need to differentiate between coupled data assimilation and coupled modelling, the study is presented as "land coupled data assimilation" however it is land data assimilation only. Please consider to re-write parts of the introduction to make this clear.

**Response:**
Thank you for your valuable feedback. We apologize for any ambiguities in our original manuscript. In response to your comment, the introduction of our manuscript has been thoroughly rewritten. Effort has been made to clearly distinguish between weakly coupled data assimilation (WCDA) and strongly coupled data assimilation (SCDA) by highlighting the differentiations between coupled modeling and coupled data assimilation. WCDA implies coupling in the forecast step, but no coupling in the analysis step. In contrast, SCDA allows coupling in both the analysis and forecast steps.

We have incorporated a more thorough description of our assimilation process and clarified that the assimilation method used in our study is referred as the WCDA system. In this study, the incorporation of GLDAS data into the E3SMv2 model consists of the analysis step and the forecast step. In the analysis step, the differences between monthly mean GLDAS data and model outputs are utilized to produce the optimal assimilation analysis. Subsequent to this, in the forecast step, the entire E3SM climate model rather than the land surface model is used as the forecast model to forecast the IC backgrounds of all components for the next assimilation window and the land reanalysis information can propagate to the other components (e.g., atmosphere and ocean) dynamically through the coupled integration of E3SM during the onemonth forecast. In general, when the coupled model is used in the forecast step while the optimal assimilation analysis is updated separately for the respective component, the assimilation approach is identified as WCDA (Penny et al., 2019; Zhang et al., 2020). Thus, the assimilation approach in this study is referred to as a WCDA system.

In the revised introduction, we first elucidate the distinctions between uncoupled data assimilation (DA) and coupled data assimilation (CDA). Uncoupled DA implies that DA is conducted using an individual component model (e.g., land surface model forced by atmospheric observations or reanalysis data rather than coupled with an atmospheric model) as the forecast model that does not consider any interactions with other components. For example, most existing reanalysis data are generated by uncoupled DA, and previous studies employ uncoupled DA that directly utilizes reanalysis data as initial conditions (ICs) to initialize decadal climate predictions (DCPs) based on coupled models (Du et al., 2012; Bellucci et al., 2013). However, such uncoupled DA often exhibits poor consistency between ICs of component models, and eventually produces low prediction skills (Balmaseda et al., 2009; Boer et al., 2016; Ardilouze et al., 2017).

To obtain balanced multi-component ICs in coupled models, recent studies focus on the development of CDA methods under the coupled modeling framework (Penny and Hamill, 2017; He et al., 2020a). The purpose of CDA is to produce balanced and coherent ICs for all components within the climate system by incorporating observational information from one or more components in the coupled model. Then CDA methods are categorized into two main types: weakly coupled data assimilation (WCDA) and strongly coupled data assimilation (SCDA).

When introducing WCDA and SCDA, we make a clear distinction between coupling in the model and coupling in the DA. Sequential DA encompasses both the analysis and the forecast steps. WCDA allows no direct influence of observations from a single component to other components in the analysis step as the cross-component background error covariances are not used, but coupling in the forecast step allows interactions across different components during the model integration (Browne et al., 2019) and propagates the observational information to other components. In contrast, SCDA utilizes cross-component background error covariances to directly assimilate the observational information from one component into all components, treating the entire Earth system model as one unified system (Penny et al., 2019). Furthermore, similar to WCDA, SCDA also allows coupling in the forecast step to propagate the observations from one component to the other components (Yoshida and Kalnay, 2018).

In response to this comment, we have revised our introduction to first elucidate the distinctions between uncoupled DA and coupled data assimilation (L41-56), and then distinguish between WCDA and SCDA by highlighting the characteristics of coupling in the model and coupled DA (L57-80). We hope that these modifications can better distinguish between uncoupled DA and CDA, as well as more effectively illustrate that the data assimilation system we developed in this study is referred to as the WCDA system.

L41-56: Much work has been devoted to initializing climate system models for practicable decadal climate predictions (DCPs). These models couple various components, such as models of the atmosphere, land surface, ocean, sea ice, and so on. Due to their much higher complexity, coupled models are often more susceptible to initial conditions (ICs) than their individual model components, underscoring the importance of dedicated data assimilation (DA) (Sakaguchi et al., 2012). The capability of DA methods is essential to incorporate available observations into the components of coupled model and produce the optimal estimate of ICs to improve DCPs. The initialization for DCPs uses uncoupled DA and coupled data assimilation (CDA) methods. Uncoupled DA performs DA under the framework of an individual component model (e.g., standalone land surface model forced by atmospheric observations or reanalysis data rather than coupled with an atmospheric model), and then the uncoupled DA analyses from different individual components are combined to form the ICs of a coupled model (Zhang et al., 2020). For example, most existing reanalysis data were produced using uncoupled DA approaches, and these reanalysis datasets are then directly used to initialize DCPs in some studies (Du et al., 2012; Bellucci et al., 2013). However, such uncoupled DA often exhibits poor consistency among the ICs of different component models, and eventually produces low prediction skills (Balmaseda et al., 2009; Boer et al., 2016; Ardilouze et al., 2017).

L57-80: To obtain balanced multi-component ICs in coupled models, recent studies focus on the development of CDA methods under the coupled modeling framework (Penny and Hamill, 2017; He et al., 2020a). The purpose of CDA is to produce balanced and coherent ICs for all components within the climate system by incorporating observational information from one or more components in the coupled model, providing great potential for improving seamless climate predictions (Dee et al., 2014). Some studies underscore the superior advantages of CDA over traditional uncoupled DA methods (Lea et al., 2015; Zhang et al., 2005). CDA methods are categorized into two main types: weakly coupled data assimilation (WCDA) and strongly coupled data assimilation (SCDA). WCDA assimilates the observations or existing reanalysis into the respective component of the coupled model and then transfers the observational information to the other components through the coupled model integration (He et al., 2020b; Zhang et al., 2020). Considering that sequential DA encompasses both the analysis and the forecast steps, WCDA allows no direct influence of observations from a single component to other components in the analysis step as the cross-component background error covariances are not used, but coupling in the forecast step allows interactions across different components during the model integration (Browne et al., 2019) and propagates the observational information to other components. In contrast, SCDA utilizes cross-component background error covariances to directly assimilate the observational information from one component into all components, treating the entire Earth system model as one unified system (Penny et al., 2019). Furthermore, similar to WCDA, SCDA also allows coupling in the forecast step to propagate the observations from one component to the other components (Yoshida and Kalnay, 2018). Several studies indicate that SCDA typically exhibits more pronounced improvements in assimilation performance relative to WCDA (Smith et al., 2015; Sluka et al., 2016). However, the application of SCDA poses substantial technical challenges, particularly in the establishment of effective cross-component background error covariances. Consequently, the majority of contemporary CDA systems still utilize the WCDA framework.

To better elucidate that our data assimilation approach in this study is referred as WCDA, we have augmented our manuscript with a more comprehensive description (L113-118, L221-228, L268-278, and L288-294) of each assimilation process with both the analysis and the forecast steps. Specifically, in the forecast step, we have emphasized that the entire E3SM climate model is utilized for forecasting, and coupling in the forecast step transfers the land reanalysis information to the other components (e.g., atmosphere and ocean) through multi-component interactions. This DA process under the coupled modeling framework is referred as the WCDA system. To distinctly differentiate our assimilation approach (WCDA) from SCDA, we have changed the terminology coupled data assimilation (CDA) to weakly coupled data assimilation (WCDA) throughout the manuscript to accurately represent our utilization of weakly coupled data assimilation. As a result, our assimilation system in this study is explicitly named the weakly coupled land data assimilation (WCLDA).

L113-118: In this WCLDA system, monthly mean anomalies of soil moisture and temperature from a global land reanalysis product are assimilated into the land component of a coupled climate model in the analysis step, and subsequently during the forecast step, the land reanalysis information incorporated into the ICs of the land component is propagated to the other components (e.g., atmosphere and ocean) through the fully coupled model integration and influences the ICs of all components for the next assimilation window.

L221-228: In the analysis step, only the land state variables are updated to the optimal analysis ($x_a^{lnd}$). Subsequently, we proceed with a one-month freely coupled integration of the E3SMv2 model during the forecast step. This integration is initialized from the optimal land ICs ($x_a^{lnd}$) along with the background fields as the ICs of other components (e.g., atmosphere and ocean). Throughout this one-month free integration, the interactions among the model components indirectly enhance the background states of these components (e.g., atmosphere and ocean) for the next assimilation window due to the more realistic land state variables. Moreover, this coupled integration also contributes to the balance between the ICs of different components.

L268-278: The incorporation of GLDAS data into the E3SMv2 model consists of the analysis step and the forecast step. In the analysis step, the differences between monthly mean GLDAS data and model outputs are calculated and utilized to produce the optimal assimilation analysis at the beginning of a one-month assimilation window. Subsequently, in the forecast step, this optimal assimilation analysis is used as the land ICs combined with the background ICs for other components to conduct one-month forecast using the E3SMv2 model. This forecast generates the backgrounds of all model components for the next assimilation window. As a result, the forecasted backgrounds for all components are influenced by the land reanalysis information incorporated into the ICs of the land component. In general, when the coupled model is used in the forecast step while the optimal assimilation analysis is updated separately for the respective component, the assimilation approach is identified as WCDA (Penny et al., 2019; Zhang et al., 2020).

L288-294: To assimilate the monthly mean GLDAS product, fully coupled integration by the E3SMv2 model is performed twice within each one-month assimilation window: first to generate the observational innovation by computing the differences between the GLDAS data and model outputs for analysis, and second to forecast the backgrounds of all components for the next assimilation window. When the fully coupled model is executed for the second one-month run, the land reanalysis information is transferred to the other components through multi-component interactions.

**Comment#2:**
As pointed out by Referee #2, the authors assimilate model derived soil moisture and temperature without taking into account the systematic differences between the two models. I fully agree with Referee #2 that the authors need to do some kind of bias correction before the assimilation step. It is not clear to me why monthly mean values are chosen and also not why you do not assimilate actual observations, please make this clear to the reader. In my opinion the authors should consider changing the experiment design and either assimilate (and evaluate) their system against actual observations or create a synthetic twin experiment study.

**Response:**
Thank you for your insightful comments. In light of your suggestions, we have now applied bias correction before assimilation and incorporated detailed explanations for our selection of monthly mean values, and assimilating land reanalysis products rather than actual observations in our revised manuscript.

Following your advice, we have modified our experiment design to add bias correction before assimilation (L168-171), and then conducted the anomaly assimilation through assimilating observed anomalies into the model. Due to the modifications of our experimental design, we have comprehensively updated all figures (Figure 3 to 10) and relevant descriptions that depict the assimilation performance with bias correction in our revised manuscript.

L168-171: In this study, we conduct the anomaly assimilation for the WCLDA system with bias correction applied to GLDAS data before assimilation. For bias correction, the difference between GLDAS data and its long-term average is calculated as anomalies and then added to the simulated model climatology.

Regarding the use of monthly mean values, we realize that our initial manuscript did not sufficiently explain this decision, which is driven by our initial interest in using data assimilation to produce initial conditions for decadal climate predictions (DCPs). Almost all initializations for DCPs in CMIP5 and CMIP6 incorporated monthly mean reanalysis data as observations (Table 1). This preference is primarily driven by two critical factors. Firstly, for decadal-scale climate predictions, assimilating data with temporal resolutions shorter than one month may introduce undesirable noise, which can adversely affect DCPs when high temporal resolution data are assimilated into the initial conditions. Hence, the prevalent practice in both CMIP5 and CMIP6 is to assimilate monthly mean data for DCPs. Secondly, the DA techniques applied in the coupled data assimilation (CDA) for initializations of decadal prediction are

generally much simpler than those used in NWPs, attributed largely to the increased complexity in coupled climate models. For examples, many initialization systems used in CMIP5 and CMIP6 adopted the simple nudging method (Table 1). Therefore, these much simpler DA approaches and much more complex coupled models do not allow the direct assimilation of actual observations. Furthermore, unlike NWPs where long-term DA cycles aren't necessary, the initialization for DCPs requires DA cycles spanning at least ten years which makes it very difficult or even impossible to assimilate complex actual observations due to the very high computational cost.

**Table 1.** Brief summaries of assimilation strategies used in CMIP5 and CMIP6 decadal prediction experiments through assimilation of reanalysis data.

| Model | Assimilation Strategies | Method | References |
|---|---|---|---|
| BCC-CSM1.1 | Ocean: assimilate the SODA reanalysis | Nudging | Xin et al., 2013 |
| CanCM4 | Atmosphere: assimilate the ERA reanalysis | Nudging | Merryfield et al., 2013 |
| CNRM-CM5 | Ocean: assimilate the NEMOVAR reanalysis | Nudging | Voldoire et al., 2014 |
| HadCM3 | Atmosphere: assimilate the ERA-40 reanalysis | Nudging | Smith et al., 2013 |
| FGOALS-g2 | Ocean: assimilate the ds285.3 reanalysis | Nudging | Wang et al., 2013 |
| EC-Earth3 | Ocean: assimilate the ORAS4 reanalysis | Nudging | Bilbao et al., 2021 |
| NorCPM1 | Ocean: assimilate the HadISST reanalysis | EnKF | Bethke et al., 2021 |
| CanE3M5 | Ocean: assimilate the ORAS5 reanalysis | Nudging | Sospedra-Alfonso et al., 2021 |

To clarify our choice of using monthly mean GLDAS reanalysis, we have incorporated detailed explanations (L245-252) in our revised manuscript.

L245-252: In contrast to decadal timescales, data signals with temporal resolutions shorter than one month can potentially introduce undesirable noise, which can adversely affect DCPs when high temporal resolution data are assimilated into the ICs. Moreover, it is very computationally demanding to assimilate complex actual observations in the initialization for DCPs that requires long-term DA cycles. Therefore, similar to most existing initialization approaches for DCPs that assimilate reanalysis data, this study describes the implementation of a data assimilation approach for initializing DCPs by assimilating monthly mean GLDAS data within the one-month assimilation window.

The key challenge we face in assimilating actual observations, particularly satellite data, arises from the lack of the observation operator within our current system. The observation operator plays a critical role in establishing the connection between the model variables and actual

observations, accounting for the discrepancies in spatial and temporal resolutions between the two datasets. It takes us one year to build this weakly coupled land data assimilation (WCLDA) system for the E3SMv2 model. Unfortunately, our current WCLDA system lacks the design of the observation operator, thereby presenting a significant obstacle to incorporating actual observational data effectively. Recognizing this limitation, we will focus on the development of the observation operator for future improvement of our WCLDA system.

To shed light on the current limitations of our WCLDA system, we have incorporated the reasons (L446-450) for its inability to assimilate actual observations in our revised manuscript. Our objective in adding these explanations is to provide readers with additional reasons behind our decision not to assimilate actual observations.

L446-450: Our current WCLDA system has some limitations such as the lack of an observation operator to integrate actual observations (e.g., satellite and station data). An observation operator is crucial in providing the linkage between the model variables and actual observations, which differ in spatial and temporal resolutions. Hence future exploration will focus on developing observation operators suitable for assimilating various satellite data, such as the AMSR-E and GRACE data.

GLDAS product generate optimal fields of land surface states and fluxes in near-real time (Rodell et al., 2004), and these reliable global GLDAS datasets are extensively utilized in weather and climate research (Chen et al., 2021; Zhang et al., 2018). In identifying an optimal long-term land surface dataset for our study, we found the GLDAS to be exceptionally suitable. Additionally, GLDAS products were also assimilated in another coupled model (FGOALS-g2), showing significant improvements in the interannual prediction skills over East Asia and Europe, as shown in previous studies by Shi et al. (2021, 2022). Therefore, we employed the advanced WCDA approach to incorporate the GLDAS monthly mean soil temperature and soil moisture into the fully coupled E3SMv2 model.

In response to your suggestion, we have expanded our analysis by further evaluating our assimilation performance against MODIS satellite observations from 2003 to 2014. We have introduced a new figure (Figure A1) in the Appendix and incorporated detailed descriptions about the assimilation performance compared with MODIS data (L348-357) in our revised manuscript. Figure A1 shows the spatial pattern of the AE index for surface soil moisture and land surface temperature between MODIS data and model simulations. For surface soil moisture, the comparison with MODIS data suggests that the majority of global regions exhibit reduced RMSE after assimilation. The reduction of RMSE is pronounced in central North America, South America, southern Africa, Australia, and Europe. However, in high-latitude areas, significant degradations are observed in northern Russia, which may be possibly related to model deficiencies in simulating the complex frozen ground and snow processes (Edwards et al., 2007; Ireson et al., 2013). Regarding land surface temperature, improved performances are evident over South America, Australia, southern Africa, and parts of Eurasia when compared to MODIS data. In contrast, some degradations appear over parts of North America and central Asia, which still require further improvement.

L348-357: We further perform an analysis of the spatial pattern of the AE index for surface soil moisture and land surface temperature between MODIS data and model simulations (Figure A1). For surface soil moisture, the comparison with MODIS data suggests that the majority of global regions exhibit reduced RMSE after assimilation. The reduction of RMSE is pronounced in central North America, South America, southern Africa, Australia, and Europe. However, in high-latitude areas, significant degradations are observed in northern Russia, which may be possibly related to model deficiencies in simulating the complex frozen ground and snow processes (Edwards et al., 2007; Ireson et al., 2013). Regarding land surface temperature, improved performances are evident over South America, Australia, southern Africa, and large parts of Eurasia when compared to MODIS data. In contrast, some degradations appear over parts of North America and central Asia, which still require further improvement.

[Figure]

**Figure A1.** Spatial distribution of the AE index for (a) surface soil moisture and (b) land surface temperature during the 2003-2014 period. The observations are derived from monthly MODIS satellite data.

Current initialization techniques consist of two main categories: full-field initialization with observed values, and anomaly initialization with observed anomalies. The optimal strategy for model initialization (full-field versus anomaly initialization) is still an active research topic (Hu et al., 2020; Polkova et al., 2019). The full-field assimilation is commonly performed to reduce the influence of systematic model biases by replacing the initial model state with the optimal available estimate of the observed state (Volpi et al., 2017). However, the model trajectory tends to drift away from the observations and revert to the model's inherent preferred state because of model deficiencies (Smith et al., 2013). This problem is partially addressed with the anomaly assimilation by assimilating the observed anomalies added to the model climatology (Carrassi et al., 2014).

We have incorporated a discussion (L161-168) to outline the advantages and disadvantages of both full-field and anomaly assimilation in our revised manuscript. This discussion also clarifies our decision to select the anomaly assimilation for the WCLDA system, emphasizing our methodology of applying bias correction to the GLDAS data before assimilation.

L161-168: Current initialization techniques are broadly classified into two categories: full-field assimilation with observed values, and anomaly assimilation with observed anomalies (Hu et al., 2020; Polkova et al., 2019). The full-field assimilation is commonly performed to reduce the influence of systematic model biases by replacing the initial model state with the optimal available estimate of the observed state (Volpi et al., 2017). However, the model trajectory tends to drift away from the observations and revert to the model's inherent preferred state because of model deficiencies (Smith et al., 2013). This problem is partially addressed with the anomaly assimilation by assimilating the observed anomalies added to the model climatology (Carrassi et al., 2014).

---

## Referee Report (RR1)

**Minor comments:**

L1: In the title you write "land weakly coupled data assimilation", while in the abstract you write "weakly coupled land data assimilation (WCLDA)" please be consistent.

L18: Skip "initial" in "With and initial.."

L59: Rephrase to skip "and so on"

L59: Skip "much higher"

L61: Skip "dedicated"

L61-63 Please rephrase, this is not what you are doing since you assimilate model data. This is what DA is used for in e.g. NWP.

L63: Add "uses both uncoupled DA…"?

L82: Not "observational information" in your case.

L104: Please revise what is done in Laloyaux et al., 2016, it is not a 3D-VAR for the whole system.

L123: Why not refer to these studies?

L124: "..or land reanalysis data"

L265: Skip "major" in "..major conclusions"

L305: "ELMv2" as defined earlier in the text

L332: What do you mean here with "independence between yi samples"?

L335: The localization methodology should be explained in more detail. Do you mean vertically or horizontally? I assume that there is no need for horizontal localization since you assimilate a model derived product which has values everywhere your model has values? Please elaborate on this.

L424: Perhaps use "constraints"?

L432-437: Should perhaps be put in Sect 2. when presenting the GLDAS product you assimilate?

L456-57: This is already mentioned, please remove

L463: Remove "to"

L496: To the best of my knowledge I don't think you have specified the value you use for the observation error covariance matrix R throughout this manuscript?

L503: Do you here use the first guess from the Assim experiment or the analysis value?

L549: You need to introduce the MODIS data you use for validation, please do so in Sect 2.

L701-2: Please rephrase as you are not assimilating observations but model data.

L726: Please rephrase "the system is conducted…"

L776-80: Please rephrase. There are globally gridded surface soil moisture products that could have been assimilated as easily as the land product you used.

---

## Author Response (AR2)

We thank Reviewer #3 for the constructive comments and suggestions, which greatly help us to improve the quality of our manuscript. We have made revisions and replied to all the comments. Our responses are shown in "Blue" color and the changes in the manuscript are shown in "Red" color.

**Response to the comments from Reviewer #3**

**Comment#1:**
L1: In the title you write "land weakly coupled data assimilation", while in the abstract you write "weakly coupled land data assimilation (WCLDA)" please be consistent.

**Response:**
Thank you for pointing out the inconsistency. We have revised both the title and the abstract to consistently use the term "weakly coupled land data assimilation (WCLDA)" throughout the manuscript.

**Comment#2:**
L18: Skip "initial" in "With an initial.."

**Response:**
We have removed "initial" to streamline our text and improve readability (L18).

**Comment#3:**
L59: Rephrase to skip "and so on"

**Response:**
We have revised this sentence to eliminate the vague reference "and so on" and provide a more explicit list of the coupled model components (L42-43).

L42-43: These models couple various components, such as models of the atmosphere, ocean, sea ice, land and river.

**Comment#4:**
L59: Skip "much higher"

**Response:**
Done

**Comment#5:**
L61: Skip "dedicated"

**Response:**
Done

**Comment#6:**
L61-63 Please rephrase, this is not what you are doing since you assimilate model data. This is what DA is used for in e.g. NWP.

**Response:**
Thank you for your valuable suggestion. We have revised this sentence (L45-46) to further emphasize the application of data assimilation in our study. Specifically, we now highlight our method of initializing climate models by assimilating reanalysis data, rather than actual observations.

L45-46: The application of DA methods is essential to incorporate reanalysis data into the components of coupled model and produce the optimal ICs to improve DCPs.

**Comment#7:**
L63: Add "uses both uncoupled DA..."?

**Response:**
Done.

**Comment#8:**
L82: Not "observational information" in your case.

**Response:**
We have revised this sentence to replace "observational information" with "reanalysis information" at Line 65.

**Comment#9:**
L104: Please revise what is done in Laloyaux et al., 2016, it is not a 3D-VAR for the whole system.

**Response:**
Thank you for pointing out the discrepancy in our initial reference to the 3DVar methodology employed in Laloyaux et al., 2016. We recognize that Laloyaux et al. (2016) implemented a 4DVar approach for the atmospheric increment and 3DVar method for the ocean increment. For accuracy, we have replaced the citation of Laloyaux et al. (2016) with Fujii et al. (2009) at Line 87, which employs a 3DVar method more aligned with the context of our discussion.

L86-87: such as the three-dimensional variational data assimilation (3DVar) (Fujii et al., 2009; Yao et al., 2021)

**Comment#10:**
L123: Why not refer to these studies?

**Response:**

Thank you for pointing out the need for specific references. We have revised this sentence to include pertinent references (He et al., 2017; Li et al., 2021; Kimmritz et al., 2018) to support our statement at Line 106.

L106: (He et al., 2017; Li et al., 2021; Kimmritz et al., 2018)

**Comment#11:**
L124: "..or land reanalysis data"

**Response:**
Done.

**Comment#12:**
L265: Skip "major" in "..major conclusions"

**Response:**
Done.

**Comment#13:**
L305: "ELMv2" as defined earlier in the text

**Response:**
Done.

**Comment#14:**
L332: What do you mean here with "independence between $y'$ samples"

**Response:**
This term "independence" implies that each $y'$ sample is selected without being influenced by the other samples in the ensemble. We have revised this sentence as "ensuring that each $y'$ sample is selected independently of the other samples in the ensemble" at Lines 205-206.

**Comment#15:**
L335: The localization methodology should be explained in more detail. Do you mean vertically or horizontally? I assume that there is no need for horizontal localization since you assimilate a model derived product which has values everywhere your model has values? Please elaborate on this.

**Response:**
We implement both horizontal and vertical localization in our data assimilation process. Despite assimilating a model-derived product that provides global coverage, we have found it necessary to implement horizontal localization to reduce sampling errors due to the finite ensemble size and to alleviate the spurious remote influence from distant grid points. Our approach to horizontal localization is to apply a distance-dependent weighting function to the

background error covariance. We employ vertical localization to limit the influence of reanalysis information on specific soil layers. More detailed descriptions of the localization methodology in our study can be found in Wang et al. (2018).

According to your comment, we have revised our manuscript to provide more detailed explanations of the localization methodology (L208-213).

L208-213: We implement both horizontal and vertical localization to reduce sampling errors due to the finite ensemble size and to alleviate the spurious remote influence from distant grid points. Our approach to horizontal localization is to apply a distance-dependent weighting function to the background error covariance. The vertical localization is employed to limit the influence of reanalysis information on specific soil layers. Please refer to Wang et al. (2018) for more detailed descriptions of the localization methodology in our study.

**Comment#16:**
L424: Perhaps use "constraints"?

**Response:**
Done.

**Comment#17:**
L432-437: Should perhaps be put in Sect 2. when presenting the GLDAS product you assimilate?

**Response:**
We have moved these sentences to the "2.2 Datasets" subsection (L159-165) of Section 2, where the GLDAS product is introduced.

**Comment#18:**
L456-57: This is already mentioned, please remove

**Response:**
Done.

**Comment#19:**
L463: Remove "to"

**Response:**
Done.

**Comment#20:**
To the best of my knowledge, I don't think you have specified the value you use for the observation error covariance matrix R throughout this manuscript?

**Response:**

Thank you for your attentive review. We have revised our manuscript to include the description of the observation error covariance (L318-319).

L318-319: The observation error covariance matrix $R$ can be determined statistically by estimating the variance and covariance of the GLDAS data.

**Comment#21:**

L503: Do you here use the first guess from the Assim experiment or the analysis value?

**Response:**

We refer to the analysis value obtained from the Assim experiment. We have revised our manuscript to explicitly state that the analysis value from Assim is used at Line 325.

**Comment#22:**

L549: You need to introduce the MODIS data you use for validation, please do so in Sect 2.

**Response:**

According to this comment, we have revised Section 2 of our manuscript to include the introduction to the MODIS data (L171-176).

L171-176: (2) MODIS is an essential instrument onboard the Terra and Aqua satellite platforms (Remer et al., 2005). The MODIS datasets provide comprehensive global observations describing atmospheric, terrestrial and oceanic conditions, including land surface temperature, vegetation indices and cloud properties (Justice et al., 2002). The MODIS products are extensively utilized for monitoring environmental changes and supporting climate change research (Gao et al., 2015; Mertes et al., 2015).

**Comment#23:**

L701-2: Please rephrase as you are not assimilating observations but model data.

**Response:**

We have revised this sentence to replace "observed anomalies" to "reanalysis anomalies" at Line 178. This terminology change underscores the use of reanalysis data rather than direct observations in our study.

**Comment#24:**

L726: Please rephrase "the system is conducted..."

**Response:**

We have revised this sentence to "Monthly mean anomalies of soil moisture and temperature from the GLDAS reanalysis are assimilated from 1980 to 2016 through the WCLDA system" for clarity (L449-450).

**Comment#25:**
L776-80: Please rephrase. There are globally gridded surface soil moisture products that could have been assimilated as easily as the land product you used.

**Response:**
Thank you for your comment. We have revised these sentences (L460-463) to emphasize our future focus on assimilating satellite-based datasets.

L460-463: Future enhancements of our WCLDA system will explore the assimilation of additional land products, particularly those derived from satellite observations. The incorporation of such satellite-based datasets is expected to further improve the performance of the WCLDA system.

**References:**
Fujii, Y., Nakaegawa, T., Matsumoto, S., Yasuda, T., Yamanaka, G., and Kamachi, M.: Coupled climate simulation by constraining ocean fields in a coupled model with ocean data, Journal of Climate, 22, 5541–5557, https://doi.org/10.1175/2009JCLI2814.1, 2009.

Gao, F., Hilker, T., Zhu, X., Anderson, M., Masek, J., Wang, P., and Yang, Y.: Fusing Landsat and MODIS data for vegetation monitoring, IEEE Geoscience and Remote Sensing Magazine, 3(3), 47–60, https://doi.org/10.1109/MGRS.2015.2434351, 2015.

He, Y., Wang, B., Liu, M., Liu, L., Yu, Y., Liu, J., Li, R., Zhang, C., Xu, S., Huang, W., Liu, Q., Wang, Y., and Li, F.: Reduction of initial shock in decadal predictions using a new initialization strategy, Geophysical Research Letters, 44(16), 8538–8547, https://doi.org/10.1002/2017GL074028, 2017.

Justice, C. O., Townshend, J. R. G., Vermote, E. F., Masuoka, E., Wolfe, R. E., Saleous, N., Roy, D. P., and Morisette, J. T.: An overview of MODIS Land data processing and product status, Remote Sensing of Environment, 83, 3–15, https://doi.org/10.1016/S0034-4257(02)00084-6, 2002.

Kimmritz, M., Counillon, F., Bitz, C. M., Massonnet, F., Bethke, I., and Gao, Y.: Optimising assimilation of sea ice concentration in an Earth system model with a multicategory sea ice model, Tellus, 70A, 1435945, https://doi.org/10.1080/16000870.2018.1435945, 2018.

Laloyaux, P., Balmaseda, M., Dee, D., Mogensen, K., and Janssen, P.: A coupled data assimilation system for climate reanalysis, Quarterly Journal of the Royal Meteorological Society, 142(694), 65-78, https://doi.org/10.1002/qj.2629, 2016.

Li, F., Wang, B., He, Y., Huang, W., Xu, S., Liu, L., Liu, J. and Li, L.: Important role of North Atlantic air–sea coupling in the interannual predictability of summer precipitation over the eastern Tibetan Plateau, Climate Dynamics, 56, 1433–1448, https://doi.org/10.1007/s00382-020-05542-6, 2021.

Mertes, C. M., Schneider, A., Sulla-Menashe, D., Tatem, A. J., and Tan, B.: Detecting change in urban areas at continental scales with MODIS data, Remote Sensing of Environment, 158, 331–347, https://doi.org/10.1016/j.rse.2014.09.023, 2015.

Remer, L. A., Kaufman, Y. J., Tanré, D., Mattoo, S., Chu, D. A., Martins, J. V., Li, R. R., Ichoku, C., Levy, R. C., Kleidman, R. G., Eck, T. F., Vermote, E., and Holben, B. N.: The MODIS

aerosol algorithm, products, and validation, Journal of the Atmospheric Sciences, 62(4), 947–973, https://doi.org/10.1175/JAS3385.1, 2005.

Wang, B., Liu, J., Liu, L., Xu, S., and Huang, W.: An approach to localization for ensemble-based data assimilation, PloS one, 13(1), e0191088, https://doi.org/10.1371/journal.pone.0191088, 2018.